# The Gain of Ordering in Online Learning

**Vasilis Kontonis**
UT Austin
vasilis@cs.utexas.edu

**Mingchen Ma**
UW-Madison
mingchen@cs.wisc.edu

**Christos Tzamos**
UW-Madison
tzamos@wisc.edu

## Abstract

We study fixed-design online learning where the learner is allowed to choose the order of the datapoints in order to minimize their regret (aka self-directed online learning). We focus on the fundamental task of online linear regression: the learner is given a dataset $X$ with $n$ examples in $d$ dimensions and at step $t$ they select a point $x_t \in X$, predict a value $\widetilde{y}_t$, and suffer loss $(\widetilde{y}_t - w^* \cdot x_t)^2$. The goal is to design algorithms that order the examples and achieve better regret than random- or worst-order online algorithms.

For an arbitrary dataset $X$, we show that, under the Exponential Time Hypothesis, no efficient algorithm can approximate the optimal (best-order) regret within a factor of $d^{1/\mathrm{poly}(\log \log d)}$.

We then show that, for structured datasets, we can bypass the above hardness result and achieve nearly optimal regret. When the examples of $X$ are drawn i.i.d. from the uniform distribution on the sphere, we present an algorithm based on the greedy heuristic of selecting "easiest" examples first that achieves a $\log d$-approximation of the optimal regret.

## 1   Introduction

In online learning [Rob51, Han57, B$^+$54, CBL06] the learner receives an example and outputs a prediction about its label. The true label of the example is then revealed and the learner suffers loss depending on the "distance" of their prediction from the true label. The goal is to minimize the total loss over all learning rounds given the knowledge of the correct labels of previous rounds. Under worst-case assumptions, where an adversary controls the sequence of examples and labels presented to the learner, a wide range of algorithms based on exponential reweighting [Vov90, LW94, FS97, Vov95, CBL06] and online convex optimization [Haz16, Ora19] have been developed.

**Beyond Worst-Case Online Learning**   A less adversarial setting is *fixed-design* (aka transductive) online learning first considered in [BDKM97] and subsequently studied in [KK05, CBS13, WHGS22]. In fixed-design online learning the pool of potential examples that the learner is going to face during the learning phase is fixed in advance and known to the learner. There are three main variants of fixed-design online learning considered in the literature (see, [BDKM97]): the *worst or adversarial order* setting, i.e., when an adversary controls the order of examples presented to the learner, the *random order*, i.e., when the examples are presented to the learner in a random order, and the *best-order or self-directed* setting [GS94], where the learner can choose the next example to predict its label at every round. Ordering examples during learning is common in practice: in the context of deep-learning, designing the order of examples when training a model is known as curriculum learning [BLCW09, HW19, WCZ21], where the focus is on finding ways to rank the training examples from "easy" to "hard", as well as using the right pacing function for introducing more difficult data. Another application is direct marketing [LL98, NL11], one common business intelligence task, which is a process of identifying likely buyers to market products accordingly. In particular, an agent must study customers' characteristics and needs, and select customers to market their products. For example, a

37th Conference on Neural Information Processing Systems (NeurIPS 2023).

streaming service or social media platform may want to learn the content preferences of customers without making too many bad recommendations. In this application, the platform chooses the order in which to present the content (from a pool of available videos) in order to minimize the "regret" (and keep the user engaged). Finally, other non-adversarial online learning variants assume that the order of examples is chosen by a teacher who knows the ground-truth and presents the examples to help the learner [GM93, Mat97, DSZ10, MSSZ22] or make regularity assumptions about the dataset of examples [RS13, JRSS15].

**Self-Directed Learning and Gain from Ordering**   In this work, we focus on the best-order or self-directed version of fixed-design online learning. We first formally define the self-directed online prediction model [GS94] and its random- and worst-order variants that we consider in this work.

**Definition 1.1** (Self-Directed Online Learning [GS94])**.** *Let $f \in \mathcal{C}$ be an unknown target concept from some concept class $\mathcal{C}$ of functions from $\mathbb{R}^d$ to $\mathbb{R}$ and let $X = \{x^{(1)}, \ldots, x^{(n)}\}$ be a subset of $n \in \mathbb{N}$ points in $\mathbb{R}^d$. The learner has access to the full set of (unlabeled) points $X$.*
*Until the labels of all examples of $X$ have been predicted:*

- *The learning algorithm picks a point $x \in X$ and makes a prediction $z \in \mathbb{R}$ about its label.*

- *The true label $f(x)$ of $x$ is revealed and the learner suffers loss $(z - f(x))^2$.*

*We say that the learner suffers $L$ loss (or regret) to label $X$ if, with probability at least $99\%$, it holds that the total loss suffered by learner over all rounds is at most $L$. In what follows, given a self-directed learner $\mathcal{A}^{\text{self}}$ we shall denote by $\mathcal{L}(\mathcal{A}^{\text{self}}, X, f)$ its total loss $L$.*

In this work, we investigate whether we can design efficient self-directed algorithms that exploit the power of ordering to improve the regret compared to the worst- and random-order settings.

**Remark 1.2** (Random-order and Worst-order Online Learning)**.** *We shall refer to the setting where the example during the training phase is picked uniformly at random (without replacement) from the unlabeled data $X$ as **random-order** learning. Moreover, we shall refer to the setting where the next example is chosen by an adversary as **worst-order** learning.*

Before we continue, we remark that we will also consider "average-case" (Bayesian) settings where the target concept $f$ is sampled from some prior distribution $F$ over concepts.

**Remark 1.3** (Average and Worst-Case Targets)**.** *In Definition 1.1, we will also consider the case where the ground-truth $f$ is drawn from some distribution over targets $F$, in which case the total cost is defined to be the average cost over $F$, i.e., $\mathcal{L}(\mathcal{A}^{\text{self}}, X, F) := \mathbf{E}_{f \sim F} \mathcal{L}(\mathcal{A}^{\text{self}}, X, f)$. In the worst case setting we will simply write $\mathcal{L}(\mathcal{A}^{\text{self}}, X) := \max_{f \in \mathcal{C}} \mathcal{L}(\mathcal{A}^{\text{self}}, X, f)$.*

To formalize the notion of how much self-directed learners "improve over worst or random settings" in a compact way, we introduce the "Gain from Ordering" (or simply gain) that is defined to be the ratio between the minimum possible total loss of an algorithm that works in the random-order setting and that of an algorithm that can order the examples.

**Definition 1.4** (Gain from Ordering)**.** *Let $f \in \mathcal{C}$ be a target concept. Let $X = \{x^{(1)}, \ldots, x^{(n)}\}$ be a dataset of points in $\mathbb{R}^d$. We define the Gain from Ordering of a self-directed learner $\mathcal{A}^{\text{self}}$ as*

$$\mathcal{G}(\mathcal{A}^{\text{self}}, X, F) := \frac{\min_{\mathcal{A}^{\text{random}}} \mathcal{L}(\mathcal{A}^{\text{random}}, X, F)}{\mathcal{L}(\mathcal{A}^{\text{self}}, X, F)}.$$

*Moreover, we denote by $\mathcal{G}^*(X, F) := \max_{\mathcal{A}^{\text{self}}} \mathcal{G}(\mathcal{A}^{\text{self}}, X, F)$ the maximum possible Gain from Ordering. Similarly, we define the Gain from Ordering in the worst-case (where the target $f$ is chosen adversarially, see Remark 1.3) by setting $\mathcal{G}(\mathcal{A}^{\text{self}}, X) := \frac{\min_{\mathcal{A}^{\text{random}}} \mathcal{L}(\mathcal{A}^{\text{random}}, X)}{\mathcal{L}(\mathcal{A}^{\text{self}}, X)}$, and $\mathcal{G}^*(X) = \max_{\mathcal{A}^{\text{self}}} \mathcal{G}(\mathcal{A}^{\text{self}}, X)$.*

We remark that in the above definition, we do not compare against worst-order algorithms as they always perform worse than the best random-order algorithm, and therefore, our results readily generalize when considering the Gain from Ordering with respect to worst-order algorithms.

Whether ordering the examples can improve the total loss suffered by the learner was first studied in the context of online classification, i.e., when the predictions of the algorithm and the ground truth labels are binary $z, f(x) \in \{\pm 1\}$. For the class of one-dimensional threshsolds over the real line, in

[GS94] it was shown that, in the best-order setting, one mistake suffices. On the other hand in the random- and worst-order settings it is known [Lit88, Lit89] that the total loss (or equivalently the number of mistakes) is $\Theta(\log n)$, where $n$ is the size of the dataset. Therefore, for thresholds on the real-line the Gain from Ordering is known to be $\Theta(\log n)$. Positive results on the Gain from Ordering exist for other concept classes as well (e.g., for monotone monomials and axis-aligned rectangles) but there are also known concepts where ordering the examples does not help, see [GS94, BDEK95].

**Online Linear Regression** In this paper, we focus on perhaps the most fundamental problem in online learning, namely online linear regression. An unknown target vector $w^* \in \mathbb{R}^d$ is picked by an adversary (or sampled from some prior distribution, see Remark 1.3). The true label of an example $x$ corresponds to $f(x) = w^* \cdot x$. In the adversarial/worst-order setting there are numerous results studying online linear regression going back to the seminal work of Widrow and Hoff [WH60, LLW91, KW97, CBLW96, Byl97, AW01, Vov01, OCCB15]. In this setting, the optimal regret is well-understood and tight (even with respect to constant factors) upper and lower bounds exist [AW01, Vov01, CBL06]. The self-directed setting is much less understood both information-theoretically, i.e., what is the best possible Gain from Ordering, and computationally, i.e., whether there exist efficient self-directed algorithms that can achieve the optimal Gain from Ordering. In this work, we aim to make progress in this direction and ask the following fundamental question.

*Is there an efficient, self-directed learning algorithm for linear regression*
*that achieves (approximately) optimal Gain from Ordering?*

## 1.1 Our Results and Techniques

Our first result is an impossibility result showing that for *unstructured datasets* it is computationally intractable to compute an ordering that approximates the optimal Gain from Ordering under the Exponential Time Hypothesis (ETH). More precisely, we show that, even for the "easier" version of the problem where the target vector is drawn uniformly at random from the unit sphere, no efficient self-directed learner can achieve better than $d^{1/\text{poly}(\log \log d)}$-approximation of the optimal Gain from Ordering.

**Theorem 1.5** (Hardness of Approximation for the Optimal Gain from Ordering)**.** *Let $X$ be an arbitrary set of $n$ unit-norm examples and denote by $\mathbb{S}^d$ the uniform distribution over the unit sphere. Under the Exponential Time Hypothesis (ETH), there is no polynomial time self-directed learner $\mathcal{A}^{\text{self}}$ such that $\mathcal{G}(\mathcal{A}^{\text{self}}, X, \mathbb{S}^d) \geq d^{-1/\log\log^c d} \, \mathcal{G}^*(X, \mathbb{S}^d)$, where $c > 0$ is some universal constant.*

Our hardness result follows by a reduction to the Densest $k$-Subgraph (DkS) problem (see Definition 3.1) that was shown in [Man17] to be ETH-hard to approximate. To go from our learning problem to the combinatorial DkS problem we perform a sequence of approximation preserving reductions. At a high-level, we first show that selecting the best order for online linear regression is equivalent to an offline geometric problem where we need to sort the examples so that the sum of the distances of every example from the subspace spanned by the previous examples in the ordering is minimized, see Definition 3.3). We then show that this problem can be further reduced to a specific edge packing problem on graphs (see Definition 3.7) by only worsening the achieved approximation guarantee by a factor of 2. Finally, we show that any $\alpha$-approximate algorithm for the $k$-packing problem yields an $O(\alpha^2)$-approximate algorithm for the Densest $k$-Subgraph problem. For more details, we refer to Section 3.

As our hardness result suggests, without any assumption over the structure of the data, there is no hope to design an efficient self-directed learner with "good" approximation guarantee – especially in high-dimensional settings, i.e., when $d$ is large. This motivates us to study self-directed learning over structured datasets. We focus on the fundamental setting where the dataset $X$ is i.i.d. drawn uniformly from the unit sphere, $S^d$. In this case, we design an algorithm based on a greedy heuristic that picks the example that is "most similar" to the examples already seen achieves a $\log d$-approximation of the optimal Gain from Ordering.

**Theorem 1.6** (Efficient Self-Directed Learner under Spherical Data)**.** *Let $X$ be a set of $n$ examples drawn i.i.d. from $\mathbb{S}^d$. There exists an efficient self-directed learner $\mathcal{A}$ such that, with probability at least 99%, it holds $\mathcal{G}(\mathcal{A}, X, \mathbb{S}^d) \geq \Omega(1/\log d) \, \mathcal{G}^*(X, \mathbb{S}^d)$. Moreover, it holds that $\mathcal{G}(\mathcal{A}, X, \mathbb{S}^d) \geq \min(\Omega(d), \Omega(n^{2/d}))$.*

We remark that, apart from presenting an efficient algorithm for approximating the Gain of Ordering, Theorem 1.6, gives the first information-theoretic bound for the optimal gain under spherical data, showing that $\mathcal{G}^*(X, \mathbb{S}^d)$ is roughly $n^{2/d}$. At a high level, the fact that the optimal gain increases slower as the dimension increases is explained by the fact that, $n$ samples from the uniform distribution on the sphere will be almost orthogonal (unless $n$ is very large). Therefore, observing the labels of a subset of them reveals little information about the remaining points. See Section 4 for more details.

Apart from showing that, under structured datasets, singnificantly improved approximation guarantees can be achieved our upper bound of Theorem 1.6 serves as a formalization of the popular greedy heuristic of picking the "easiest examples first" used in curriculum learning ([BLCW09]) for linear regression. In particular, assuming that we have already observed the labels of the points $x^{(1)}, \ldots, x^{(i)}$ we pick the example that is closest to the subspace spanned by $x^{(1)}, \ldots, x^{(i)}$, see Algorithm 1.

As an extension, we also study the self-directed ReLU regression problem, where the concept class is $\mathcal{C} = \{\text{Relu}(w^* \cdot x) \mid w^* \in \mathbb{R}^d\}$. ReLU regression is a very popular non-convex optimization task that has recently received significant attention both in online and offline settings, [KS09, Sol17, GKKT17, GKK, DGK$^+$20, DKKZ20, DKZ20, FCG20, DKTZ22, DKR23], due to the fact that ReLU is a very common activation in deep learning models. In Section 5, we present an efficient algorithm that achieves Gain of Ordering roughly $\min(d, n^{2/d}/\log d + \log n/\log d)$, see Theorem 5.1, assuming a "warm-start" labeled example. We observe that, even though ReLU regression is typically a harder task than linear regression, the Gain of Ordering that we achieve is larger (by a term of $\log n/\log d$) than that of linear regression. At a high-level this interesting phenomenon has to do with the the ReLU being constant on a large region of the space: a fact that the self-directed learner can exploit to improve its gain. It is an interesting direction for future research to further investigate the properties of the activation and the dataset that allow for improved gains by self directed learners.

## 1.2 Related Work

Related to the setting of self-directed learning is active learning [CAL94], where the learner has access to a large pool of unlabeled examples and chooses the "most informative" to ask for their labels. The goal is to find a classifier with good generalization while minimizing the number of label queries. There is a long line of research on active linear classification in the distribution-specific setting (e.g., under the uniform distribution on the unit sphere) [DKM05, Han11, BU16]. We remark that our goal of minimizing the number of mistakes is orthogonal to that of active learning: at a high-level, our algorithms pick the examples for which the current hypothesis is most confident ("easiest examples") while in active learning one typically asks for the labels of the "hardest examples", e.g., those with the smallest margin with respect to the current guess (see, e.g., [ABHU15, ABHZ16, ZSA20]).

In deep learning, stochastic gradient descent typically trains models by considering the examples in a random order. In the influential work of [BLCW09] the authors proposed curriculum learning: training machine learning models in a "meaningful order" – from easy examples to harder ones. There is a long line of research (see the surveys [HW19, WCZ21, SIRS22] and references therein) giving empirical evidence that curriculum learning provides significant benefits in convergence speed and generalization over training with random order. Our results provide theoretical evidence that ordering the examples from easier to harder significantly reduces the mistakes made by the learner.

There has been a long line of work studying online linear regression. We first review the results in the realizable setting, which means $y(x) = w^* \cdot x$. In this case, the problem is also called the adaptive filtering problem by [KWH06]. Online gradient descent was shown to be a minimax optimal method by [KWH06, CBLW96]. These works show if $\|x\| \leq B$ and $\|w^*\| \leq W$, then online gradient descent achieves a regret bound $O(B^2 W^2)$. A matching lower bound $\Omega(B^2 W^2)$ was given by [CBLW96] using a dataset that contains only one example. In the adversarial setting, online gradient descent and exponentiated gradient were studied by [CBLW96, KW97]. Given the knowledge of $B$, by suitable tuning the learning rate, the two algorithms achieve a regret bound $O(L(w^*))$, which grows linearly with $T$ in the worst case. Furthermore, if the algorithms know an error bound $E$ for $L(w^*)$ and a bound $W$ for $\|w^*\|$, then these algorithms can achieve a regret bound $O(\sqrt{E})$, which is $O(\sqrt{T})$ in the worst case. These regret bounds cannot be further improved for such types of algorithms. On the other hand, assuming $y(x) \leq Y$ and $\|x\| \leq B$ for any $x \in X$, [Vov01, AW01] obtained a regret bound of $O\left(\|w^*\|^2 + dY^2 \log(TX^2/d)\right)$ using a so called online nonlinear ridge regression method.

A matching lower bound of $\Omega(\|w^*\|^2 + Y^2 d \log T)$ were given by [CL06, Vov01]. Recently better lower bound of $\Omega(dY^2 \log T)$ were given by [BKM+15, GGHS19]. Another interesting setting is the stochastic setting studied by [OMP21], where $y(x) = w^* \cdot x + \xi$, $\xi$ is a zero-mean sub-gaussian noise with variance $\sigma^2$. In particular, in such a setting, the label $y$ can be unbounded. Ouhamma et al. showed that the online (nonlinear) ridge regression with high probability has a regret bound of $O(d\sigma^2 \log T \log \log T)$.

## 2 Notation and Preliminaries

In this section, we introduce the notations we will use in the paper. Let $X \subseteq \mathbb{R}^d$ be the set of $n$ examples Denote by $w^* \in \mathbb{R}^d$ the target vector that labels each $x \in X$ by $y(x) = w^* \cdot x$. Let $\mathcal{A}$ be a self-directed learner. For $i \in [n]$, we use random vector $x^{(i)} \in \mathbb{R}^d$ to denote the example in $X$ that is selected by $\mathcal{A}$ in the $i$-th round. We denote by $L_i$ be subspace spanned by examples $x^{(1)}, \ldots, x^{(i)}$ and $L_i^\perp$ the subspace that is orthogonal to $L_i$. For every $v \in \mathbb{R}^d$ and for every subspace $L \subseteq \mathbb{R}^d$, we denote by $v_L = \text{proj}_L(v)$, the projection of $v$ onto $L$. Furthermore, we will use $S^d$ to denote the unit sphere in $\mathbb{R}^d$ and use $\mathbb{S}^d$ to denote the uniform distribution over $S^d$

## 3 Optimal Gain from Ordering under Arbitrary Data is Hard

In this section, we show that self-directed linear regression, in general, is hard to approximate, even when $X \subseteq S^d$ and $w^*$ is drawn uniformly from $S^d$.

In the rest of this section, we will give a high-level overview of the proof of Theorem 1.5. The full proof of results in this section can be found in Appendix A. The key idea is that if we have a good efficient self-directed learner, then we can obtain an efficient algorithm that approximately solves the $k$-densest subgraph problem (D$k$S), which has been shown hard to approximate by [Man17].

**Definition 3.1** (Densest $k$-Subgraph Problem (D$k$S))**.** *Let $G = (V, E)$ be an undirected graph with $n$ vertices and $m$ edges, and let $k \in [n]$. The goal is to find a subset of $k$ vertices $S$ such that the edge density $\rho(S) := |E(S)|/\binom{|S|}{2}$ is maximized, where $E(S)$ denotes the set of all edges among the vertices in $S$. We define* opt *to be the maximum density over all possible subsets of $k$ vertices. Given $G$ and $k$, an $\alpha$-approximate algorithm for DkS problem outputs a subset of $k$ vertices $S$ such that $\alpha\rho(S) \geq$ opt *in polynomial time.*

**The Sequential Spanning Problem (SSP)**   To begin with, we observe that every self-directed learner has two parts, selecting examples and making predictions. The first observation, which is stated as Lemma 3.2, is that with a prior distribution of $w^*$, it is easy to obtain a Bayesian optimal prediction in each round.

**Lemma 3.2.** *Let $X = \{x^{(1)}, \ldots, x^{(n)}\} \subseteq \mathbb{R}^d$ be a set of $n$ examples and $w^*$ be a target vector drawn uniformly from $S^d$ that labels $y^{(i)} = w^* \cdot x^{(i)}$ for each $x^{(i)} \in X$. Given given any set of labeled examples $(x^{(1)}, y^{(1)}), \ldots, (x^{(i-1)}, y^{(i-1)})$, denote by $L_{i-1}$ the subspace spanned by $x^{(1)}, \ldots, x^{(i-1)}$ and $w^*_{L_{i-1}}$ the projection of $w^*$ onto $L_{i-1}$. Let $\mathcal{A}$ be a self-directed learner, denote by $\hat{y}^{(i)}$ be the prediction of $\mathcal{A}$ for the next example $x^{(i)}$, then we have $\mathbf{E}_{w^*}\left((\hat{y}^{(i)} - y^{(i)})^2 \mid w^*_{L_{i-1}}\right) \geq \mathbf{E}_{w^*}\left((w^*_{L_{i-1}^\perp} \cdot x^{(i)})^2 \mid w^*_{L_{i-1}}\right)$. Furthermore, the inequality holds for equality if $\hat{y}^{(i)} = w^*_{L_{i-1}} \cdot x^{(i)}$.*

With the Bayesian optimal prediction, the problem becomes how to select a good ordering. We show in Proposition 3.4, the problem can be equivalent to formulate as the following Sequential Spanning Problem.

**Definition 3.3** ($k$-Sequential Spanning Problem ($k$-SSP))**.** *Let $X = \{x^{(1)}, \ldots, x^{(m)}\}$ on $\mathbb{R}^d$ be a set of $m$ points with unit norm and let $k \in \mathbb{N}$. Consider an ordered sequence $\sigma$ of $k$ points of $X$, i.e., $\sigma = x^{(i_1)}, x^{(i_2)}, \ldots, x^{(i_k)}$ and define $L_j$ to be the sub-space spanned by the first $j$-elements of $\sigma$ i.e., $L_j = \text{span}(x^{(i_1)}, \ldots x^{(i_j)}) = \text{span}(x^{(\sigma(1))}, \ldots x^{(\sigma(j))})$ and $V_0 = \emptyset$. Define the following cost, called spanning cost,*

$$C(\sigma, k) = \sum_{i=1}^{k} \|\text{proj}_{L_{i-1}^\perp} x^{(\sigma(i))}\|_2^2.$$

*We define* opt *to be the minimum cost over all sequences of $k$-elements of $X$. Given $X$ and $k$, an $\alpha$-approximate algorithm for $k$-SSP Problem outputs an ordered sequence of $k$ points from $X$ so that $C(\sigma, k) \leq \alpha$ opt in polynomial time.*

*We shall refer to the special case of the problem where $k = m$ simply as the Sequential Spanning Problem.*

**Proposition 3.4** (From SSP to Self-Directed Learning)**.** *Let $X = \{x^{(1)}, \ldots, x^{(m)}\} \subseteq \mathbb{R}^d$ be a set of examples. For every self-directed linear regression algorithm $\mathcal{A}$ over $X$, with expected learning loss $\mathcal{L}(\mathcal{A}, \mathbb{S}^d)$, we can use it to get a randomized algorithm $\mathcal{A}'$ for SSP over $X$ with expected cost $\mathbf{E}(C(\sigma, m)) \leq d\, \mathcal{L}(\mathcal{A}, \mathbb{S}^d)$. Moreover, given a randomized algorithm $\mathcal{A}'$ for SSP over $X$, we can get a self-directed linear regression algorithm $\mathcal{A}$ over $X$, with expected learning loss $\mathcal{L}(\mathcal{A}, \mathbb{S}^d) = \mathbf{E}(C(\sigma, m))/d$. In particular, the construction can be done efficiently.*

With the intuition above, we only need to show it is ETH-hard to approximate the SSP. The key technical result we obtain is the following proposition.

**Proposition 3.5** (From D$k$S to SSP)**.** *For every function $\alpha(d, m) : R^+ \times R^+ \to R^+$, if there is an $\alpha(d, m)$-approximate algorithm for the SSP, then there is a $64\alpha^2(n^3, n^3)$-approximate algorithm for the DkS problem.*

As a direct corollary of Proposition 3.5, we obtain the computational hardness of SSP.

**Corollary 3.6** (ETH-hardness of SSP)**.** *Assuming the Exponential Time Hypothesis (ETH) is true, then there is no algorithm that outputs an $d^{1/\log\log^c d}$-approximate solution to the SSP in $\mathrm{poly}(d, m)$ time, where $c > 0$ is a universal constant.*

*Proof of Corollary 3.6.* By Corollary 1.3 in [Man17] if the Exponential Time Hypothesis (ETH) is true, then there is no polynomial time $n^{1/\log\log^c n}$-approximate algorithm for $k$-densest subgraph problem. By Proposition 3.5, if there is a polynomial time $d^{1/\mathrm{poly}\log\log d}$-approximate algorithm for SSP, we can obtain an efficient $n^{1/\log\log^c n}$ approximate algorithm for $k$-densest subgraph problem. Thus, if ETH is true it is hard to approximate SSP within a $d^{1/\mathrm{poly}\log\log d}$ factor in polynomial time. $\qquad\square$

In the rest of the section, we introduce the high-level of the proof of Proposition 3.5 by breaking it down into two steps.

**From $k$-Edge Packing to SSP**     In the first step, we introduce an intermediate problem called $k$-Edge Packing. We want to show if we can approximate SSP efficiently then we can approximate $k$-Edge Packing efficiently.

**Definition 3.7** ($k$-Edge Packing)**.** *Let $G = (V, E)$ be an undirected graph with $n$ vertices and $m$ edges, and let $k \in [n]$. The goal is to find a subset of $k$ edges $S$ such that the number of vertices covered by $S$, $|V(S)|$ is minimized, where $V(S)$ denotes the set of endpoints of edges in $S$. We define opt to be the minimum number of vertices covered by any possible subsets of $k$ edges. Given $G$ and $k$, an $\alpha$-approximate algorithm for the $k$-edge packing problem outputs a subset of $k$ edges $S$ such that $V(S) \leq \alpha$opt in polynomial time.*

The key idea in this step is to show Lemma 3.8, if we can approximate $k$-SSP efficiently, then we can approximate $k$-edge packing efficiently.

**Lemma 3.8.** *For every function $\alpha : R^+ \times R^+ \to R^+$, if there is an $\alpha(d, m)$-approxiate algorithm for $k$-SSP problem, then there is a $2\alpha(n, m)$-approximate algorithm for $k$-edge packing problem*

To build a connection between a geometric problem and a combinatorial problem, we have the following construction. Given a graph $G = (V, E)$ with $n$ vertice and $m$ edges, we will map every edge $(u, v)$ to a sparse $n$-dimensional vector $x_{uv}$ such that the only non-zero entries are $x_u = 1$, $x_v = -1$. In this way, we obtain a dataset $X$ as the input of $\mathcal{A}$, the approximate algorithm for $k$-SSP. Intuitively, if we select $k$ edges in $G$ that are disjoint, then no matter how to order the corresponding vectors of these $k$ edges, the spanning cost is $\Omega(k)$. But if we select $k$ edges that form a clique, the corresponding vectors span a subspace of dimension $O(\sqrt{k})$, so it is easy to order these vectors to get a very small spanning cost. With such an intuition, the key structure result we use here is that if a solution to the $k$-SSP problem has a sufficiently small cost, then the corresponding edges to these $k$

vectors must be sufficiently connected to each other and thus cover a sufficiently small number of vertices.

However, now we are only able to approximate the $k$-edge packing problem with an algorithm that approximately solves the $k$-SSP problem. To finish this step, we should also be able to use an algorithm for SSP as a subroutine to solve the $k$-SSP problem. We show this is possible in Lemma 3.9.

**Lemma 3.9.** *Assume that an algorithm for SSP in $d$ dimensions exists that finds an $\alpha(d, m)$-approximate solution in $\mathrm{poly}(d, m)$ time for some function $\alpha : R^+ \times R^+ \to R^+$. Then an algorithm that finds a $4\alpha(m(d + m), m(d + m))$-approximate solution for $k$-SSP for every value of $k$ in $\mathrm{poly}(d, m)$ time exists.*

The intuition here is that if the dataset $X$ is in general position (every set of $d$ examples are linearly independent) then SSP is actually a $d$-SSP. Given any dataset $X$ with $m$ points, we are able to map $X$ to a dataset $X'$ in $f(k, d, m) > d$ dimension that is in the general position. Such a map is done by making multiple copies for each example, lifting them to high dimension and adding tiny structured noise to each of them. We will show that such a transformation well preserves the information in the original dataset so that if we run $\mathcal{A}$ over $X'$, we can extract a good approximate solution to the original $k$-SSP from the first $f(k, d, m)$ terms of the output solution.

**From D$k$S to $k$-Edge Packing**    So far we have shown how to use an algorithm for SSP to solve the $k$-edge packing problem. Our final step is to show Lemma 3.10, which implies that if we can approximately solve the $k$-edge packing problem then we can also solve the $k$ densest subgraph problem.

**Lemma 3.10** ($k$-edge packing and D$k$S)**.** *For $\alpha > 0$, if there is an $\alpha$-approximate algorithm for $k$-edge packing problem, then there is an $\alpha^2$-approximate algorithm for D$k$S problem.*

Notice that $k$-edge packing is seeking $k$ edges that cover as few vertices as possible, which can be thought as a dual problem of D$k$S. If we are able to find the largest number $f(k)$ such that $f(k)$ edges can cover at most $k$ vertices, then these $f(k)$ edges induce the densest subgraph with $k$-vertices. The idea behind Lemma 3.10 is that we can approximately find such $f(k)$ using an approximate algorithm for $k$-edge packing and thus can approximately find the $k$-densest subgraph.

With the above three lemmas, we can prove Proposition 3.5.

**Proof of Proposition 3.5**    Assume we have an $\alpha(d, m)$-approximate algorithm for SSP problem, then by Lemma 3.9, we get an efficient $4\alpha((d + m)m, (d + m)m)$-algorithm for $k$-SSP problem. By Lemma 3.8, we get an efficient $8\alpha((m+n)m, (m+n)m)$-approximate algorithm for $k$-edge packing problem. Since $(m + n)m \le n^3$ always holds, we get an efficient $8\alpha(n^3, n^3)$-approximate algorithm for $k$-edge packing problem. By Lemma 3.10, this gives us an efficient $64\alpha^2(n^3, n^3)$-approximate algorithm for $k$-densest subgraph problem.

# 4    A $O(\log d)$-Approximation for Spherical Data

Theorem 1.5, suggests that without any assumption on the structure of the data, it is hard to obtain an efficient self-directed learners that approximates the best possible improvement over learning in random order. This motivates us to study the learning problem over datasets with natural structures. In this section, we consider perhaps the most natural setting, where the dataset $X$ is drawn i.i.d. from $\mathbb{S}^d$. Our main result in this section shows a simple greedy heuristic, Algorithm 1, which selects the "easiest" example in each round, and has a nearly (i.e., off by a $\log d$-factor) optimal gain of ordering.

We first bound the Gain of Ordering achieved by the greedy heuristic described in Algorithm 1.

**Proposition 4.1.** *Let $X$ be a set of $n \ge \mathrm{poly}(d)$ examples drawn i.i.d. from $\mathbb{S}^d$ and let $\mathcal{A}$ denote Algorithm 1, then over the randomness of the dataset $X$, in expectation, $\mathcal{G}(\mathcal{A}, X, \mathbb{S}^d)^{-1} \le O(1/d + n^{-2/d})$.*

**Algorithm 1** SELFDIRECTEDLINEARREGRESSION($X$)

---

Randomly partition $X$ into $d$ subsets $X_1, \ldots, X_d$ such that $X_i$ contains $O(n/(d-i)^2)$ examples.
**for** $i = 1, \ldots, d$ **do**
    Set $L_{i-1} = \mathrm{span}\{x^{(1)}, \ldots, x^{(i-1)}\}$.
    Find $w^{(i)} \in L_{i-1}$ consistent with $(x^{(1)}, y^{(1)}), \ldots, (x^{(i-1)}, y^{(i-1)})$.
    Select $x^{(i)} = \mathrm{argmin}_{x \in X_i} \left\| \mathrm{proj}_{L_{i-1}}(x) \right\|$.
    Predict $\hat{y}^{(i)} = w^{(i)} \cdot x^{(i)}$ and receive $y^{(i)}$.
Label all unlabeled examples in $X$ using $w^{(d)}$.

---

We start with some intuition for Algorithm 1. Since the observed labels $y^{(i)}$ are consistent with $w^*$, after selecting examples $x^{(1)}, \ldots, x^{(i)}$, our guess $w^{(i)}$ at step $i$ coincides with the projection $w^*_{L_i}$ of $w^*$ onto the subspace $L_i$ spanned by these examples. If we choose any $x^{(i+1)}$ and predict $w^*_{L_i} \cdot x^{(i+1)}$, then we will pay $\ell_i^2 = (w^*_{L_i^\perp} \cdot x^{(i+1)})^2 = (w^*_{L_i^\perp} \cdot x^{(i+1)}_{L_i^\perp})^2$. Since we make a random partition of the data in advance, we are able to show that in expectation $\ell_i^2 / (\left\| w^*_{L_i^\perp} \right\|^2 \left\| x^{(i+1)}_{L_i^\perp} \right\|^2) = 1/(d-i)$.

This suggests the cost we pay in each round is proportional to $\left\| x^{(i+1)}_{L_i^\perp} \right\|^2$ and we should choose the example that is closest to $L_i$ greedily. Our key technical lemma shows that in the $i+1$-th round, for a greedily chosen example and a randomly chosen example, the expected ratio of the loss is $O(n^{-2/(d-i)})$.

We next show that the optimal Gain from Ordering can only be $O(\log d)$ times larger than that of Algorithm 1.

**Proposition 4.2** (Bounding the Optimal Gain). *Let $X \subseteq S^d$ be a set of $n > \mathrm{poly}(d)$ examples drawn i.i.d. uniformly from $S^d$. For every $\delta \in (3/n, 1)$, with probability at least 99%, $\mathcal{G}^*(X, S^d)^{-1} \geq \Omega(1/d + \delta n^{-(2+2\delta)/d})$.*

Recall that in Section 3, we showed that if $w^* \sim S^d$, then the expected learning cost is proportional to the optimal spanning cost of a sequence of examples. This implies analyzing the best learner is equivalent to analyzing the sequence of examples in $X$ with the smallest spanning cost. The key of our proof is to show that the distribution of $S^d$ is very concentrated and unless the size of $X$ is very large there is no sequence of examples whose spanning cost is much smaller than the average spanning cost.

As a direct corollary, we are able to show Algorithm 1 has a gaining that is at most $O(1/\log d)\mathcal{G}^*(X, S^d)$, which is an exponential improvement of the hardness of approximation obtained by Theorem 1.5.

**Proof of Theorem 1.6** By Proposition 4.1 and Proposition 4.2, we know that with probability at least 99%, $\mathcal{G}(\mathcal{A}, X, S^d)^{-1} \leq 1/d + n^{-2/d}$ and $\mathcal{G}^*(X, S^d)^{-1} \geq 1/d + \delta n^{-(2+2\delta)/d}$. Here we ignore the hidden constant within the bound obtained from Proposition 4.1 and Proposition 4.2 because it will only add some multiplicative constant factor in our final result. It remains to tune the parameter $\delta$ to show that $\frac{1/d + n^{-2/d}}{1/d + \delta n^{-(2+2\delta)/d}}$ is at most $O(\log d)$. To do this, we write $s^{d/2}$ and we consider different ranges for $s$. We consider the following two cases.

Case 1: If $s \geq d$, then we have we have $1/d + n^{-2/d} \leq 2/d \leq 2(1/d + \delta n^{-(2+2\delta)/d})$, which implies $\mathcal{G}(\mathcal{A}, X, S^d) \geq \Omega(1)\mathcal{G}^*(X, S^d)$.

Case 2: If $s < d$, then $n^{-2/d} \geq 1/d$ and $1/d + \delta n^{-(2+2\delta)/d} \geq \delta n^{-(2+2\delta)/d}$. This implies $\frac{1/d + n^{-2/d}}{1/d + \delta n^{-(2+2\delta)/d}} \leq \frac{2}{\delta} n^{2\delta/d} = \frac{2}{\delta} s^\delta$. We set $\delta = 1/\log s$ and we obtain that $\frac{1}{\delta} s^\delta = s^{1/\log s} \log s = O(\log s) \leq O(\log d)$. This implies that $\mathcal{G}(\mathcal{A}, X, S^d) \geq \Omega(1/\log d)\mathcal{G}^*(X, S^d)$.

**Remark 4.3.** *We remark that although the statement of Proposition 4.1 and Proposition 4.2 are about the average performance of the algorithm, the same results also hold under the worst-case setting because, with a dataset from a spherical distribution, the learning cost of $\mathcal{A}^{\mathrm{random}}$ under*

*these two settings are asymptotically the same in expectation. A detailed discussion about this and the proof of Proposition 4.1 and Proposition 4.2 are deferred to Appendix B.*

**Remark 4.4.** *Although according to Theorem 1.6, Algorithm 1 approximates the optimal gaining within a $O(\log d)$ factor, when $n \leq s^d$ for some constant $s$ that doesn't depend on $d$ or $n \geq d^{d/2}$ Algorithm 1 approximates the optimal gaining within a constant factor and thus is nearly optimal.*

## 5 Self-Directed ReLU Regression

Finally, we study the problem of self-directed ReLU regression, which shares a similar spirit to the one of self-directed linear regression. According to Proposition 4.1, given a dataset $X$ that is drawn from $\mathbb{S}^d$, we are able to design an efficient learner with gaining $\mathcal{G}(\mathcal{A}, X) \geq \min(\Omega(d), \Omega(n^{2/d}))$ if each $x$ is labeled by $y(x) = w^* \cdot x$. In fact, the bound of $\min(\Omega(d), \Omega(n^{2/d}))$ can also be obtained by the following Algorithm 2, when each $x$ is labeled by a ReLU function $\mathrm{Relu}(w^* \cdot x)$. However, an interesting phenomenon we found is that if we give Algorithm 2 some reference example $(x^{(0)}, y^{(0)})$ as a warm start such that $\theta(x^{(0)}, w^*) = \theta_0 < \pi/2$, then we are able to improve the gaining to $\min(\Omega(d), \Omega(n^{2/d}/\log d + \log n/\log d))$, which is a huge improvement when the dimension of the problem is large.

---

**Algorithm 2** SELFDIRECTEDRELUREGRESSION($X$)

---

Let $(x^{(0)}, y^{(0)})$ be a pair of reference example such that $\theta(x^{(0)}, w^*) = \theta_0 \leq \pi/2$.
(Assume $(x^{(0)}, y^{(0)}) = (0, 0)$ if there is no such a warm start)
Randomly partition $X$ into $d$ subsets $X_1, \ldots, X_d$ such that $X_i$ contains $n/d$ examples.
**for** $i = 1, \ldots, d$ **do**
    Set $L_{i-1} = \mathrm{span}\{x^{(0)}, \ldots, x^{(i-1)}\}$, where $x^{(j)}$ is example that has been selected with positive label for $j \in \{0, \ldots, i-1\}$.
    Find $w^{(i)} \in L_{i-1}$ consistent with $(x^{(0)}, y^{(0)}), \ldots, (x^{(i-1)}, y^{(i-1)})$.
    Keep selecting $x \in X_i$ such that $w^{(i)} \cdot x \in \mathrm{argmin}_{x' \in X_i} w^{(i)} \cdot x'$ and predict $\mathrm{Relu}(w^{(i)} \cdot x)$
until we see some $(x^{(i)}, y^{(i)})$ such that $y^{(i)} > 0$.
Label all unlabeled examples in $X$ using $w^{(d+1)}$.

---

**Theorem 5.1.** *Let $X \subseteq \mathbb{R}^d$ be a set of $n \geq \mathrm{poly}(d)$ examples drawn i.i.d. from $\mathbb{S}^d$. Let $\mathcal{G}(\mathcal{A}, X)$ be the gain from ordering of Algorithm 2 for the self-directed ReLU regression problem over $X$. Then*

$$\mathcal{G}(\mathcal{A}, X)^{-1} \leq O\left(\frac{1}{d}\right) + \begin{cases} \min\{O(\frac{\tan^2 \theta_0 \log d}{d}), O(\frac{\log d}{n^{2/d}})\} & \text{if } \frac{n\theta_0}{4\pi d \log d} > \exp(\frac{d}{8}), \\ \min\{O(\frac{\tan^2 \theta_0 \log d}{\log(n\theta_0)}), O(\frac{\log d}{n^{2/d}})\} & \text{if } 1 \leq \frac{n\theta_0}{4\pi d \log d} \leq \exp(\frac{d}{8}) \\ O(\frac{1}{n}) & \text{if } \frac{n\theta_0}{4\pi d \log d} < 1. \end{cases}$$

The main difference between ReLU regression and linear regression is that in ReLU regression, with a good warm start, we are able to start from the region that is labeled 0 by the target to seek the decision boundary. Before we see a positive example, we pay nothing. The first time we see a positive example $x$, we pay $(w^* \cdot x)^2$. The key technical lemma, Lemma 5.2 shows that as we keep selecting the most "negative" example with respect to our current guess, the first positive example we see must have a very small margin with respect to $w^*$. In this way, Algorithm 2 learns $w^*$ with a very small cost. We refer to Appendix C for more details and proofs.

**Lemma 5.2** (Geometry Technical Lemma). *Let $w^* \in S^d$ be a target vector and let $w \in S^d$ be an arbitrary vector such that $\theta = \theta(w^*, w) < \pi/2$. Denote by $C = \{x \in S^d \mid w^* x \leq 0, w \cdot x \geq 0\}$. For every $a, b \in (0, 1)$, denote by $K_a := \{x \in C \mid w \cdot x \geq a \sin \theta\}$ and $K_{a,b} := \{x \in C \mid w \cdot x \geq a \sin \theta, w^* x \leq -b \sin \theta\}$. Let $x$ be a point uniformly drawn from $S^d$. There is some absolute constant $c > 1$ such that if $a/b \geq c$ then $\mathbf{Pr}\left(x \in K_{a,b} \mid x \in K_a\right) \leq 2 \exp\left(-\frac{d}{3(1-a^2)}(b^2 + 2ab \cos \theta)\right)$.*

## 6 Conclusion, Limitations, and Broader Impact

In this work, we study the self-directed learning problem for linear regression. Our work presents novel results both computational and information-theoretic. Our first result shows that approxi-

mating the optimal self-directed regret within a $d^{1/\mathrm{poly}\log\log d}$ factor is ETH-hard under arbitrary datasets. Our second result yields a novel characterization of the Gain of Ordering for data that are uniformly distributed on the sphere and gives an efficient approximation algorithm that bypasses the aforementioned hardness result and achieves close to optimal regret by exploiting the structure of the data. A limitation of our work is that our algorithms can currently handle linear (or ReLU regression). Moreover, our presented results assume that the labels are realizable – even in this fundamental setting, nothing was known prior to our work. Generalizing our results to other concept classes and investigating the effect of adding noise to the labels are natural questions for future investigation. Designing robust self-directed learning algorithms for broader concept classes under broader distributional assumptions are interesting direction for future work.

## 7   Acknowledgements

This work was supported by the NSF Award CCF-2144298 (CAREER).

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
