# A  Missing Proofs from Section 3

## A.1  Equivalence of Learning a Random Target and Sequential Spanning Problem

In this subsection, we show that when $w^*$ is drawn from $\mathbb{S}^d$, we can equivalently formulate the sell-directed linear regression problem as the Sequential Spanning Problem (SSP). In particular, we will prove Proposition 3.4, which we restated here.

**Proposition A.1** (Proposition 3.4). *Let $X = \{x^{(1)}, \ldots, x^{(m)}\} \subseteq \mathbb{R}^d$ be a set of examples. For every self-directed linear regression algorithm $\mathcal{A}$ over $X$, with expected learning loss $\mathcal{L}(\mathcal{A}, \mathbb{S}^d)$, we can use it to get a randomized algorithm $\mathcal{A}'$ for SSP over $X$ with expected cost $\mathbf{E}(C(\sigma, m)) \leq d\, \mathcal{L}(\mathcal{A}, \mathbb{S}^d)$. Moreover, given a randomized algorithm $\mathcal{A}'$ for SSP over $X$, we can get a self-directed linear regression algorithm $\mathcal{A}$ over $X$, with expected learning loss $\mathcal{L}(\mathcal{A}, \mathbb{S}^d) = \mathbf{E}(C(\sigma, m))/d$.*

To start with, we show that when $w^* \sim \mathbb{S}^d$, we can describe the Bayes optimal prediction. In particular, we show that given the points $x^{(1)}, \ldots, x^{(i-1)}$ the best possible prediction for the label $y^{(i)}$ of some point $x^{(i)}$ is $w^*_{L_i} \cdot x^{(i)}$, where $L_i$ is the subspace spanned by $x^{(1)}, \ldots, x^{(i-1)}$. We remark that the vector $w^*_{L_i}$ can be efficiently estimated by the samples $(x^{(1)}, y^{(1)} = w^* \cdot x^{(1)}), \ldots (x^{(i-1)}, y^{(i-1)} = w^* \cdot x^{(i-1)})$ by solving a linear system. We show the following lemma.

**Lemma A.2** (Lemma 3.2). *Let $X = \{x^{(1)}, \ldots, x^{(n)}\} \subseteq \mathbb{R}^d$ be a set of $n$ examples and $w^*$ be a target vector drawn uniformly from $S^d$ that labels $y^{(i)} = w^* \cdot x^{(i)}$ for each $x^{(i)} \in X$. Given any set of labeled examples $(x^{(1)}, y^{(1)}), \ldots, (x^{(i-1)}, y^{(i-1)})$, denote by $L_{i-1}$ the subspace spanned by $x^{(1)}, \ldots, x^{(i-1)}$ and $w^*_{L_{i-1}}$ the projection of $w^*$ onto $L_{i-1}$. Let $\mathcal{A}$ be a self-directed learner, denote by $\hat{y}^{(i)}$ be the prediction of $\mathcal{A}$ for the next example $x^{(i)}$, then we have $\mathbf{E}_{w^*}\left((\hat{y}^{(i)} - y^{(i)})^2 \mid w^*_{L_{i-1}}\right) \geq \mathbf{E}_{w^*}\left((w^*_{L^{\perp}_{i-1}} \cdot x^{(i)})^2 \mid w^*_{L_{i-1}}\right)$. Furthermore, the inequality holds with equality if $\hat{y}^{(i)} = w^*_{L_{i-1}} \cdot x^{(i)}$.*

*Proof of Lemma 3.2.* We first observe that $y^{(i)} = w^* \cdot x^{(i)} = w^*_{L_{i-1}} \cdot x^{(i)} + w^*_{L^{\perp}_{i-1}} \cdot x^{(i)}$. We have

$$
\begin{aligned}
&\mathbf{E}_{w^*}\left((\hat{y}^{(i)} - y^{(i)})^2 \mid w^*_{L_{i-1}}\right) \\
&= \mathbf{E}_{w^*}\left((\hat{y}^{(i)} - w^*_{L_{i-1}} \cdot x^{(i)} - w^*_{L^{\perp}_{i-1}} \cdot x^{(i)})^2 \mid w^*_{L_{i-1}}\right) \\
&= \mathbf{E}_{w^*}\left((\hat{y}^{(i)} - w^*_{L_{i-1}} \cdot x^{(i)})^2 \mid w^*_{L_{i-1}}\right) - 2\,\mathbf{E}_{w^*}\left((\hat{y}^{(i)} - w^*_{L_{i-1}} \cdot x^{(i)})(w^*_{L^{\perp}_{i-1}} \cdot x^{(i)}) \mid w^*_{L_{i-1}}\right) \\
&\quad + \mathbf{E}_{w^*}((w^*_{L^{\perp}_{i-1}} \cdot x^{(i)})^2 \mid w^*_{L_{i-1}}) \\
&\geq \mathbf{E}_{w^*}\left((w^*_{L^{\perp}_{i-1}} \cdot x^{(i)})^2 \mid w^*_{L_{i-1}}\right),
\end{aligned}
$$

where the inequality holds because $\mathbf{E}_{w^*}\left(w^*_{L^{\perp}_{i-1}} \cdot x^{(i)} \mid w^*_{L_{i-1}}\right) = 0$, as $w^*$ is drawn uniformly from the unit sphere. $\qquad\square$

With the Bayes optimal prediction rule, we are able to connect the self-directed learning problem with SSP directly via the following Lemma. We show that the cost of any self-directed algorithm that picks a sequence of $d$ points $x^{(1)}, \ldots, x^{(d)}$ of a dataset $X$ is at least $(1/d)\sum_{i=1}^d \|x^{(i)}_{L^{\perp}_{i-1}}\|^2$, i.e., at every step the algorithm pays for the (squared) length of $x^{(i)}$ that is not contained in the subspace spanned by its first $i-1$ choices $x^{(1)}, \ldots, x^{(i-1)}$.

**Lemma A.3.** *Let $w^*$ be a uniform random vector over $S^d$. Let $\{x^{(1)}, \ldots, x^{(m)}\}$ be a sequence of points in $\mathbb{R}^d$. Denote by $L_i$ the subspace spanned by $x^{(1)}, \ldots, x^{(i)}$, for $i \in [d]$. Every self-directed learner $\mathcal{A}$ that learns $w^*$ with (random) sequence $\{x^{(1)}, \ldots, x^{(d)}\}$ has an expected cost $\mathcal{L}(\mathcal{A}, \mathbb{S}^d) \geq \frac{1}{d}\mathbf{E}\left(\sum_{i=1}^d \left\|(x^{(i)})_{L^{\perp}_{i-1}}\right\|^2\right)$. Furthermore, the inequality holds for equality if $\mathcal{A}$ always uses the prediction rule in Lemma 3.2.*

*Proof of Lemma A.3.* We may assume that $m = d$ and random points $\{x_1, \ldots, x_d\}$ are linearly independent. By Lemma 3.2, we know that the optimal hypothesis we use to predict in each round is $w^*_{L_{i-1}}$. Based on this observation, we obtain that

$$\mathcal{L}(\mathcal{A}, \mathbb{S}^d) = \mathbf{E} \sum_{i=1}^d (\hat{y}^{(i)} - y^{(i)})^2 = \sum_{i=1}^d \mathop{\mathbf{E}}_{w^*} \left( (\hat{y}^{(i)} - y^{(i)})^2 \mid w^*_{L_{i-1}} \right) \geq \sum_{i=1}^d \mathop{\mathbf{E}}_{w^*} \left( (w^*_{L^\perp_{i-1}} \cdot x^{(i)})^2 \mid w^*_{L_{i-1}} \right)$$

$$= \sum_{i=1}^d \mathop{\mathbf{E}}_{w^*} \left( (w^*_{L^\perp_{i-1}} \cdot (x^{(i)})_{L^\perp_{i-1}})^2 \mid w^*_{L_{i-1}} \right)$$

$$= \sum_{i=1}^d \mathop{\mathbf{E}}_{w^*} \left( \left\| w^*_{L^\perp_{i-1}} \right\|^2 \left\| (x^{(i)})_{L^\perp_{i-1}} \right\|^2 (\frac{w^*_{L^\perp_{i-1}}}{\left\| w^*_{L^\perp_{i-1}} \right\|} \cdot \frac{(x^{(i)})_{L^\perp_{i-1}}}{\left\| (x^{(i)})_{L^\perp_{i-1}} \right\|})^2 \mid w^*_{L_{i-1}} \right)$$

$$= \mathbf{E} \sum_{i=1}^d \frac{d - i + 1}{d} \frac{1}{d - i + 1} \left\| (x^{(i)})_{L^\perp_{i-1}} \right\|^2 = \mathbf{E} \sum_{i=1}^d \frac{1}{d} \left\| (x^{(i)})_{L^\perp_{i-1}} \right\|^2 .$$

Here, in the second last equality, we use the fact that for any fixed $k$ dimensional subspace, the expected square of the norm of $w^*$ (which is drawn uniformly from the unit sphere) projected onto that subspace is $k/d$.

$\square$

The proof of Proposition 3.4 can be obtained as a direct corollary of Lemma A.3.

*Proof of Proposition 3.4.* We first show how to construct a randomized algorithm $\mathcal{A}'$ for SSP based on a self-directed learner $\mathcal{A}$. $\mathcal{A}'$ works as follows. We draw some $w^* \sim \mathbb{S}^d$ run $\mathcal{A}$ over $X$ to learn $w^*$. $\mathcal{A}'$ outputs $\sigma$, the order of the examples we select during the learning process. By Lemma A.3, we know that $\mathbf{E}\, C(\sigma, m) \leq d\mathcal{L}(\mathcal{A}, \mathbb{S}^d)$.

Next, we show that given a randomized algorithm $\mathcal{A}'$ for SSP over $X$, we can efficiently construct a self-directed learner. We run $\mathcal{A}'$ over $X$ to obtain an order $\sigma$ of examples in $X$. Learner $\mathcal{A}$ selects examples according to $\sigma$ and make a prediction according to Lemma A.3. By Lemma A.3, we know that $\mathcal{L}(\mathcal{A}, \mathbb{S}^d) = \mathbf{E}\, C(\sigma, m)/d$.

$\square$

## A.2 Approximating $k$-SSP via Approximating SSP

In this subsection, we show that if we can compute an approximate solution to SSP over $X$ efficiently, then we can also compute an approximate solution to $k$-SSP over $X$ efficiently for every $k$. In this section, we will use $\alpha(d, m) : \mathbb{N} \times \mathbb{N} \mapsto \mathbb{R}^+$ to denote the approximate ratio of an approximate algorithm. Formally, we will prove Lemma 3.9, which we restate again here.

**Lemma A.4** (Lemma 3.9). *Assume that an algorithm for SSP in $d$ dimensions exists that finds an $\alpha(d, m)$-approximate solution in $\mathrm{poly}(d, m)$ time. Then an algorithm that finds a $4\alpha(m(d + m)), m(d + m))$-approximate solution for $k$-SSP for every value of $k$ in $\mathrm{poly}(d, m)$ time exists.*

Let $X \subseteq \mathbb{R}^d$ be a set of $m$ points. We say $X$ is in general position if any subset of $k$ points of $X$ is linearly independent. The proof of Lemma 3.9 can be broken down into two steps.

In the first step, we claim that given any set of examples $X$, we are able to put them to general position efficiently while preserving the SSP-cost of every permutation of $X$ within a factor of 2.

**Claim A.5.** *Given $X \subseteq \mathbb{R}^d$, a set of $m$ points, we can efficiently construct a map $f : X \to \mathbb{R}^{d+m}$ such that $f(X)$ is in general position. Furthermore, for every sequence of points $\sigma = (x^{(1)}, \ldots, x^{(m)})$, denote by $f(\sigma) = (f(x^{(1)}), \ldots, f(x^{(m)}))$, then $C(\sigma, k) \leq C(f(\sigma), k) \leq 2C(\sigma, k)$.*

*Proof of Claim.* Assume $X = \{x^{(1)}, \ldots, x^{(m)}\}$. Without loss of generality, we assume $\min_{i \in [m]} \left\| x^{(i)} \right\| = 1$. The mapping $f$ is defined as follows. We map each $x^{(i)}$ to a vector $f(x^{(i)}) = x^{(i)'} + \epsilon e_{d+i}$ in $\mathbb{R}^{d+m}$ for some $\epsilon \leq 1/k^2$. Here, $(x^{(i)'})_j = x^{(i)}_j$ if $j \leq d$, $(x^{(i)'})_j = 0$

otherwise and $e_{d+i}$ is $d+i$-th standard basis vector in $\mathbb{R}^{d+m}$. It is easy to see that $f(X)$ is linearly independent and thus $f(X)$ is in general position. It remains to show $f$ preserve the cost of an ordered sequence of points. Let $\sigma$ be any sequence of points. Without loss of generality, we assume that $\sigma = (x^{(1)}, \ldots, x^{(m)})$. To simplify the notation, we denote by $f(L_i)$ the subspace spanned by $\{f(x^{(1)}), \ldots, f(x^{(i)})\}$. Based on our construction, we have decomposition $f(L_i) = L'_i \oplus E_i$, where $L'_i$ is the subspace spanned by $x^{(1)'}, \ldots, x^{(i)'}$ and $E_i$ is the subspace spanned by $e_{d+1}, \ldots, e_{d+i}$. This implies that

$$\text{proj}_{f(L_i)^\perp} f(x^{(i+1)'}) = \text{proj}_{(L'_i)^\perp} x^{(i+1)'} + \epsilon e_{d+i+1}.$$

Thus, we obtain that

$$C(\sigma, k) \le C(f(\sigma), k) = C(\sigma, k) + k\epsilon^2 \le 2C(\sigma, k),$$

where the last inequality follows by $C(\sigma, k) \ge 1$ and $k\epsilon^2 \le 1$.

$\diamond$

In the second step, we show that given any set of examples $X$ in general position, we are able to construct an efficient approximate algorithm for $k$-SSP over $X$ via an efficient approximate algorithm for SSP over $X$.

**Claim A.6.** *Let $X \subseteq S^d$ be a set of $m$ points in general position. Let $\mathcal{A}$ be an $\alpha(d, m)$-approximate algorithm for geometry latency problem. We can efficiently use $\mathcal{A}$ to obtain a $2\alpha(dk, dm)$-approximate solution to the $k$-SSP over $X$.*

*Proof of Claim.* Without loss of generality, we assume $k < d$. Otherwise, if $k \ge d$, we can run $\mathcal{A}$ over $X$ directly and pick the first $k$ points in order in the output solution. This solution is an $\alpha(d, m)$-approximate solution. So in the rest of the proof, we assume $k < d$.

Let $X \subseteq S^d$ be a set of $m$ points in general position. We first construct a set of $dm$ examples $X'$ in $\mathbb{R}^{dk}$ as an input to $\mathcal{A}$. For $i \in [m]$, we denote by $x^{(i)'}$ the following vector in $\mathbb{R}^{dk}$. $(x^{(i)'})_j = x_j^{(i)}$ if $j \le d$, $(x^{(i)'})_j = 0$ otherwise. For $i \in [dk]$, denote by $e_i$ the $i$th standard basis in $\mathbb{R}^{dk}$. For each $x^{(i)} \in X$, we create $d$ examples $x^{(ij)}$ for $j \in [d]$ such that $x^{(ij)} = x^{(i)'} + \epsilon\xi^{(ij)}$, where $\xi^{(ij)}$ is a random vector drawn uniformly from the unit sphere, in the subspace spanned by $e_{d+1}, \ldots, e_{kd}$ and $\epsilon > 0$ is a tiny number we will determine later.

Next, we run $\mathcal{A}$ over the new dataset $X'$ and get a solution $\sigma' = x^{(i_1 j_1)}, x^{(i_2 j_2)}, \ldots$ to the SSP over $X'$. We construct a solution $\sigma$ in the following way. For $\ell \in [dm]$, we write $x^{(i_\ell j_\ell)} = x^{(i_\ell)'} + \epsilon\xi^{(i_\ell j_\ell)}$. We go through $\sigma'$ and every time we see a point $x^{(i_\ell j_\ell)}$ constructed from a new $x^{(i_\ell)'}$. We append $x^{i_\ell}$ to our solution $\sigma$. We output $\sigma$ if we have appended $k$ points to $\sigma$. Notice that for every $i, j$, $x^{(i)'}$ and $\xi^{(ij)}$ are orthogonal to each other. So every time a point $x^{(ij)}$ constructed from some new $x^{(i)}$ is included in $\sigma'$, we pay at least $\left\| \text{proj}_{L_{i-1}^\perp} x^{(i)} \right\|^2$, where $L_{i-1}$ is the subspace spanned by the previous $i-1$ examples in $X$ that we have used. This implies that

$$C(\sigma, k) \le C(\sigma', dm).$$

Denote by $\sigma^*$ the optimal solution to the $k$-SSP over $X$. We will show $\sigma$ is a good approximate solution by constructing another solution $\bar{\sigma}$ to the SSP over $X'$ such that $C(\bar{\sigma}, dm)$ lower bounds $C(\sigma^*, k)$ within a factor of 2. We construct $\bar{\sigma}$ in two stages. Assume $\sigma^* = (x^{(i_1)}, \ldots, x^{(i_k)})$. In the first stage, for $\ell \in [k]$, we append $(x^{(i_\ell 1)}, \ldots, x^{(i_\ell d)})$ to $\bar{\sigma}$. In the second stage, we append the remaining points in $X'$ to $\bar{\sigma}$ with arbitrary order. We first bound the cost in the first stage. Use the fact that for every $i, j$, $x^{(i)'}$ and $\xi^{(ij)}$ are orthogonal to each other again. It is not hard to see the cost in the first stage is at most $C(\sigma^*, k) + dk\epsilon^2$. As $C(\sigma^*, k) \ge 1$, we have the cost in the first stage is at most $2C(\sigma^*, k)$ by choosing $\epsilon^2 \le 1/(dk)$.

It remains to show that with probability 1, the cost of the second stage is 0. Since in the first stage, we added $dk$ points to $\bar{\sigma}$, it is sufficient to show that with probability 1, these $dk$ examples are linearly independent. Suppose we have added $(x^{(i_1 1)}, \ldots, x^{(i_1 d)}), \ldots, (x^{(i_{\ell-1} 1)}, \ldots, x^{(i_{\ell-1} d)}), (x^{(i_\ell 1)}, \ldots, x^{(i_\ell j)})$ to $\bar{\sigma}$ and these $d(\ell-1)+j$ points are linearly independent. We show that with probability 1 if we add $x^{(i_\ell, j+1)}$ to $\bar{\sigma}$ they are still

linearly independent. Since $X$ is in general position, we know that $x^{(i_1)}, \ldots, x^{(i_\ell)}$ are linearly independent. This implies if $x^{(i_\ell, j+1)}$ is in the subspace spanned by the previous $d(\ell - 1) + j$ points then $x^{(i_\ell, j+1)}$ is in the subspace spanned by $(x^{(i_\ell 1)}, \ldots, x^{(i_\ell j)})$. Again, we use the fact that for every $i, j$, $x^{(i)'}$ and $\xi^{(ij)}$ are orthogonal to each other. This implies that $\xi^{(i_\ell, j+1)}$ is in the subspace spanned by $(\xi^{(i_\ell 1)}, \ldots, \xi^{(i_\ell j)})$. This probability is 0 because the noise vectors are drawn from a $d(k-1)$-dimensional sphere uniformly. This implies that with probability 1 the points we added in the first stage are linearly independent.

So, we get $C(\bar{\sigma}, k) \leq 2C(\sigma^*, k)$. To finish the proof, we have

$$C(\sigma, k) \leq C(\sigma', dm) \leq \alpha(dk, dm)C(\bar{\sigma}, k) \leq 2\alpha(dk, dm)C(\sigma^*, k).$$

So we construct a $2\alpha(dk, dm)$-approximate solution $\sigma$ to the $k$-SSP over $X$ efficiently.

$\diamond$

*Proof of Lemma 3.9.* Let $\mathcal{A}$ be an $\alpha(d, m)$-approximate algorithm for SSP. Denote by opt the optimum of the $k$-SSP over $X$. Assume that opt is achieved by some sequence $\sigma^*$ Since $X$ may not be in general position, we use Claim A.5 to obtain a dataset $f(X)$ that is in general position. Let $f(\sigma)$ be an $\alpha$-approximate solution to the $k$-SSP over $f(X)$. Then $\sigma$ is a $2\alpha$-approxiate solution to the $k$-SSP over $X$. This is because

$$C(\sigma, k) \leq C(f(\sigma), k) \leq \alpha C(f(\sigma^*), k) \leq 2\alpha C(\sigma^*, k).$$

Now we have a set of $m$ points $f(X)$ in $\mathbb{R}^{d+m}$ that is in general position. By Claim A.6, we know that we can use $\mathcal{A}$ to obtain a $2\alpha((d+m)k, (d+m)m)$-approximate solution to the $k$-SSP over $f(X)$. Thus, we conclude that we can efficiently find a $4\alpha((d+m)k, (d+m)m)$-approximate solution to the $k$-SSP over $X$.

$\square$

## A.3   Approximating $k$-edge Packing via Approximating $k$-SSP

In this subsection, we show that given an approximate algorithm for $k$-SSP, we can use it to design an efficient approximate algorithm for $k$-edge packing.

**Lemma A.7** (Lemma 3.8). *If there is an $\alpha(d, m)$-approxiate algorithm for $k$-SSP problem, then there is a $2\alpha(n, m)$-approximate algorithm for $k$-edge packing problem*

*Proof of Lemma 3.8.* Let $G = (V, E)$ be an instance of $k$-edge packing problem, and let $A$ be an efficient algorithm for $k$-SSP. We will show we can use $\mathcal{A}$ to construct an efficient algorithm that outputs a $2\alpha$-aprroximate solution to the $k$-edge packing problem. To do this, we first construct an instance of $k$-SSP based on $G$. Recall that $G$ contains $n$ vertices and $m$ edges. For every edge $e = (u, w), u < w$, we construct a vector $v_e \in \mathbb{R}^n$ such that $(v_e)_u = 1, (v_e)_w = -1$ and $(v_e)_s = 0$ for $s \notin \{u, w\}$. We run $\mathcal{A}$ over $X = \{v_e\}_{e \in E}$ and obtain a sequence $\sigma = (v_{e_1}, \ldots, v_{e_k})$ of $k$ points in $X$. Our goal is to show $S = \{e_1, \ldots, e_k\}$ is a $2\alpha$-approximate solution to the $k$-edge packing problem.

We first lower bound $C(\sigma, k)$. Recall that $C(X, \sigma) = \sum_{i=1}^{k} \|\text{proj}_{L_{i-1}^{\perp}} v_{e_i}\|_2^2$, where $L_{i-1}$ is the subspace spanned by $v_{e_1}, \ldots, v_{e_{i-1}}$. Now we only consider the cost for every $e_i$ such that when we add $e_i$ to $\{e_1, \ldots, e_{i-1}\}$, we introduce new vertices to the solution. There are two cases we will consider. In the first case, adding $e_i$ will introduce two new vertices $u, w$. This implies that $(v_{e_j})_u = (v_{e_j})_w = 0$ for every $j \leq i - 1$. Thus $v_{e_i} \perp L_{i-1}$ and the cost of picking $v_{e_i}$ is 2. In the second case, adding $e_i$ will only introduce one new vertex $u$. This implies that $(v_{e_j})_u = 0$ for every $j \leq i - 1$ and thus we will pay at least 1 for picking $v_{e_i}$. This gives the following lower bound $C(\sigma, k) \geq |V(S)|$, where $V(S)$ is the set of vertices covered by $S$.

Next, let $S^*$ be the optimal solution to the $k$-edge packing problem. We will use $S^*$ to construct a solution $\sigma^*$ to the $k$-SSP such that $C(\sigma^*, k) \leq 2\text{opt}$, where opt is the optimum of the $k$-edge packing problem. We first decompose $S^*$ into disjoint connected components $C_1, \ldots, C_\ell$. The order $\sigma^*$ is constructed in two stages.

In the first stage, we start with an arbitrary edge $e_1 \in C_1$ and add $v_{e_1}$ to $\sigma^*$. Next, we keep adding $v_{e_i} \in C_1$ so that we only introduce one new vertex by adding $e_i$ to $\sigma^*$. If adding one edge from $e_1$

will not introduce new vertices in $C_1$, then we start adding vectors whose corresponding edges are from $C_2$ in the same way. The first stage finishes when we will not introduce any new vertex by adding any vector to $S^*$. In the second stage, we add vectors corresponding to the remaining edges with an arbitrary order.

Notice that there are at most opt vertices. This implies that in the first stage, we add at most opt vectors and each vector charges us at most 2. So, the cost in the first stage is at most 2opt. To show $C(\sigma^*, k) \leq 2$opt, we will show the cost is 0 in the second stage. Notice that after the first stage, the edges we added to $\sigma^*$ construct $\ell$ spanning trees. Thus, adding any new edge will create a cycle. Assume $u_1, u_2, \ldots, u_\ell$ to be an arbitrary path in $G$ such that $v\{v_{(u_i, u_{i+1})} \mid i \in [\ell - 1]\}$ has been added to $\sigma^*$. Furthermore, we assume that we are adding $v_{(u_1, u_\ell)}$ to $\sigma^*$ in the second stage. Since $v_{(u_1, u_\ell)} = \sum_{i=1}^{\ell-1} v_{(u_i, u_{i+1})}$, we know the cost of adding $v_{(u_1, u_\ell)}$ is 0. Thus, the cost for the second stage is 0 and we obtain that $C(\sigma^*, k) \leq 2$opt.

Finally, we conclude that $S$ is a $2\alpha(n, m)$-approximate solution to the $k$-edge packing problem because

$$|V(S)| \leq C(\sigma, k) \leq \alpha(n, m)C(\sigma^*, k) \leq 2\alpha(n, m)\text{opt}.$$

Thus, we constructed an efficient $2\alpha(n, m)$-approximate algorithm for the $k$-edge packing problem using $\mathcal{A}$. $\square$

## A.4    Approximating D$k$S via Approximating $k$-edge Packing

In this subsection, we show if we can compute an approximate solution to $k$-edge packing efficiently then we can approximate D$k$S.

**Lemma A.8** (Lemma 3.10)**.** *If there is an $\alpha$-approximate algorithm for $k$-edge packing problem, then there is an $\alpha^2$-approximate algorithm for D$k$S problem.*

*Proof of Lemma 3.10.* For every $k \in [n]$, we define $\rho_k$ to be the optimal density for the D$k$S problem over graph $G$. To prove the lemma, we first show the following claim.

**Claim A.9.** $\rho_k \geq \rho_{k+1}$ for every $k \in [n]$.

*Proof of Claim.* Assume there is some $k$ such that $\rho_k < \rho_{k+1}$ instead. Let $S_{k+1}$ be a set of $k + 1$ vertices such that $\rho(S_{k+1}) = \rho_{k+1}$. This implies

$$|E(S_{k+1})| = \rho_{k+1}\frac{k(k+1)}{2} = \rho_{k+1}\frac{k(k-1)}{2} + \rho_{k+1}k.$$

Notice that the minimum degree of a vertex in $S_{k+1}$ is at most $\rho_{k+1}k$, otherwise $S_{k+1}$ has more than $\rho_{k+1}\binom{k+1}{2}$ edges. This implies that if we delete the vertex with minimum degree from $S_{k+1}$, we get a subgraph of $G$ that contains $k$ vertices and $\rho_{k+1}\binom{k}{2} > \rho_k\binom{k}{2}$. This contradicts the definition of $\rho_k$. $\diamond$

Now we show how to use Claim A.9 to prove Lemma 3.10. Denote by $\mathcal{A}(k)$ an $\alpha$-approximate algorithm for $k$-edge packing problem, we will show we can run $\mathcal{A}$ for $m$ times to output an $\alpha^2$-approximate solution to the D$k$S problem. Our algorithm works as follows. For $i \in [m]$, we run $\mathcal{A}(i)$ to solve an $i$-edge packing problem over graph $G$. Denote by $i^*$ the largest number such that using the solution output by $\mathcal{A}(i^*)$, we cover $k$ vertices. Denote by $S^*$ the corresponding set of vertices. We show that $S^*$ gives an $\alpha^2$-approximate solution to the D$k$S problem.

We observe that for every $i \in [m]$, if the minimum number of vertices that can be covered by $i$ edges is $j$, then $i = \rho_j\binom{j}{2}$. Since $\mathcal{A}$ is an $\alpha$-approximate algorithm for $i^*$-edge packing problem, we know that there is some $k' \in [k/\alpha, k]$ such that

$$i^* = \rho_{k'}\binom{k'}{2} \geq \rho_k\frac{k'(k'-1)}{2} \geq \frac{1}{\alpha^2}\rho_k\binom{k}{2},$$

where the first inequality follows by Claim A.9 and the second inequality follows by $k' \geq k/\alpha$. This implies that $\rho(S^*) \geq \rho_k/\alpha^2$ and thus we get an efficient $\alpha^2$-approximate algorithm for D$k$S problem. $\square$

# B  Missing Proofs from Section 4

## B.1  Labeling a Random Set of Examples

Since in this section, we are focusing on the cost of labeling a random set, which is a slightly different task from labeling a fixed set in Section 3, before presenting the missing proofs in Section 4, we first reexplain the notations and the benchmark to avoid confusion.

We start with the worst-case setting. Let $\mathcal{C}$ be a concept class of function from $\mathbb{R}^n \to \mathbb{R}$ and let $f \in \mathcal{C}$ be the unknown target concept. Let $\mathcal{A}$ be a self-directed learner and let $X$ be a random set of examples drawn from some distribution $D$. Let random variable $L(\mathcal{A}, X, f)$ be the loss suffered by $\mathcal{A}$ during the learning process, where the randomness comes from the dataset $X$ and the internal randomness of $\mathcal{A}$. We say $\mathcal{A}$ labels dataset $X$ with loss $L$ if with probability 99% , $L(\mathcal{A}, X, f) \leq L$. We will use $\mathcal{L}(\mathcal{A}, X, f) \in \mathbb{R}$ to denote such a bound of loss $L$. Furthermore, $\mathcal{L}(\mathcal{A}, X) = \max_{f \in \mathcal{C}} \mathcal{L}(\mathcal{A}, X, f)$.

A slightly easier setting is when $f$ is drawn from a prior distribution $F$. This average-case setting is a very natural relaxation of the worst-case setting. In this setting, for a fixed set of examples $X$, we use $\mathcal{L}(\mathcal{A}, X, F) = \mathbf{E}_{\mathcal{A}, f} L(\mathcal{A}, X, f)$ to denote the expected loss suffer by $\mathcal{A}$, where the randomness comes from $f$ and $\mathcal{A}$. When $X$ is drawn from a distribution $D$, $\mathcal{L}(\mathcal{A}, X, F)$ is a random variable of $X$. In this setting, we are more interested in if with probability at least 99%, we can draw a dataset $X$ from $D$ such that $\mathcal{A}$ can approximate the best learner to learn $f \sim F$ over $X$.

Finally, we will consider the gain from ordering for labeling a random dataset $X$. We observe that for the linear regression problem when $X \sim D$ and the size of $X$ is large, a random sequence of $d$ examples from $X$ is roughly a sequence of i.i.d. examples. Based on this observation, instead of competing with the performance of a learner $\mathcal{A}^{\mathrm{random}}$ that selects a random order for every realization of $X$, we choose to compete with the average performance of $\mathcal{A}^{\mathrm{random}}$. This is a reasonable benchmark and will not make our analysis over complicated due to some extreme cases. Formally, for the average-case setting, the benchmark we want to compete with is $\min_{\mathcal{A}^{\mathrm{random}}} \mathcal{L}(\mathcal{A}^{\mathrm{random}}, X) \in \mathbb{R}$. And the gain from the ordering of a learner $\mathcal{A}$ is a random variable

$$\mathcal{G}(\mathcal{A}, X, F) = \frac{\min_{\mathcal{A}^{\mathrm{random}}} \mathbf{E}_{X \sim D} \mathcal{L}(\mathcal{A}^{\mathrm{random}}, X, F)}{\mathcal{L}(\mathcal{A}, X, F)},$$

where the randomness only comes from $X$. For the worst-case setting, the benchmark we want to compete with is $\min_{\mathcal{A}^{\mathrm{random}}} \max_{f \in \mathcal{C}} \mathbf{E}_{X \sim D} L(\mathcal{A}^{\mathrm{random}}, X, f) \in \mathbb{R}$ and the gain from the ordering of a learner $\mathcal{A}$ is a positive number

$$\mathcal{G}(\mathcal{A}, X) = \frac{\min_{\mathcal{A}^{\mathrm{random}}} \max_{f \in \mathcal{C}} \mathbf{E}_{X \sim D} L(\mathcal{A}^{\mathrm{random}}, X, f)}{\mathcal{L}(\mathcal{A}, X)}.$$

## B.2  Learning in Random Order

To understand the gain of ordering examples, we need to first understand the performance of the best self-directed learner that learns with a random order. We will show that $\min_{\mathcal{A}^{\mathrm{random}}} \mathbf{E}_{X \sim D} \mathcal{L}(\mathcal{A}^{\mathrm{random}}, X, \mathbb{S}^d), \min_{\mathcal{A}^{\mathrm{random}}} \max_{f \in \mathcal{C}} \mathbf{E}_{X \sim D} L(\mathcal{A}^{\mathrm{random}}, X, f) \in \Theta(1)$, which can be obtained immediately by Lemma B.2 and Lemma B.3. To start with, we present the following probability lemma that will be used multiple times in this section.

**Lemma B.1.** *Let $L \subseteq \mathbb{R}^d$ be a $k$-dimensional subspace, where $k \leq d$. Let $w$ be a random vector sampled uniformly from $S^{d-1}$. Then the random variable $\|\mathrm{proj}_L(w)\|^2 \sim \mathrm{Beta}(k/2, (d-k)/2)$.*

*Proof of Lemma B.1.* Without loss of generality, we assume $L$ is the subspace spanned by the first $k$ basis vectors. Then the random variable $\|\mathrm{proj}_L(w)\|^2 = \sum_{i=1}^{k} w_i^2$ is equivalent to the the random variable $\sum_{i=1}^{k} (x^i)^2 / \sum_{i=1}^{d} (x^i)^2$, where for $i \in [d]$, $x^i$ is independently drawn from a standard normal distribution $N(0, 1)$. The distribution of $\sum_{i=1}^{k} (x^i)^2 / \sum_{i=1}^{d} (x^i)^2$ is $\mathrm{Beta}(k/2, (d-k)/2)$. $\square$

Given the above probability lemma, we are able to compute the following information-theoretic lower bound for any learner that learns from a random order.

**Lemma B.2** (Information Theoretic Lower Bound for Learning from A Random Order). *Let $w^* \sim \mathbb{S}^d$ be a target vector, and let $X$ be a set of $n \geq \text{poly}(d)$ examples drawn i.i.d. from $\mathbb{S}^d$. Denote by $\mathcal{A}$ a learner that learns $w^*$ with a random order of $X$. In expectation, we have $\mathbf{E}_{X \sim D} \mathcal{L}(\mathcal{A}, X, \mathbb{S}^d) \geq \frac{1}{2}$.*

*Proof of Lemma B.2.* By Lemma A.3, we know that in expectation, $\mathcal{L}(\mathcal{A}, X, \mathbb{S}^d) \geq$ $\mathbf{E} \sum_{i=1}^d \frac{1}{d} \left\| (x^{(i)})_{L_{i-1}^{\perp}} \right\|^2$, where $x^{(i)}$ is the $i$-th example selected in the random order. Since $X$ is drawn i.i.d. from $\mathbb{S}^d$ and the order we select is uniform at random, we know from Lemma B.1 that $\left\| (x^{(i)})_{L_{i-1}^{\perp}} \right\|^2 \sim \text{Beta}((d-i+1)/2, (i-1)/2)$ and has an expectation $(d-i+1)/d$. Sum these expectations together, we obtain that $\mathcal{L}(\mathcal{A}, X, \mathbb{S}^d) \geq \frac{1}{2}$. $\qquad\square$

On the other hand, given any target $w^*$, a learner that selects a random order of examples also learns $w^*$ with an expected cost of $O(1)$.

**Lemma B.3** (Upper Bound the Cost of Learning from A Random Order). *There is a self-directed learner $\mathcal{A}$ that learns an arbitrary $w^* \in S^d$ with an expected cost $\mathbf{E}_{X, \mathcal{A}} L(\mathcal{A}, X, w^*) \leq \frac{1}{2}$ by selecting a random order of examples of $X$, where $X$ is a set of $n \geq \text{poly}(d)$ examples drawn i.i.d. from $\mathbb{S}^d$.*

*Proof of Lemma B.3.* The self-directed learner $\mathcal{A}$ simply selects a random order $x^{(1)}, \ldots, x^{(n)}$ from $X$ and predicts $x^{(i+1)}$ with $w_{L_i}^* \cdot x^{(i+1)}$. We know that with probability 1, $x^{(1)}, \ldots, x^{(d)}$ are linearly independent. Since $X$ is a set of examples drawn i.i.d. from $\mathbb{S}^d$ and $\mathcal{A}$ selects a random order, we have

$$
\begin{aligned}
\mathop{\mathbf{E}}_{X, \mathcal{A}} L(\mathcal{A}, X, w^*) = \mathbf{E} \sum_{i=1}^d (y^{(i)} - \hat{y}^{(i)})^2 &= \sum_{i=1}^d \mathop{\mathbf{E}}_{L_{i-1}, x^{(i)}} \left( (y^{(i)} - \hat{y}^{(i)})^2 \mid x^{(i)} \right) \\
&= \sum_{i=1}^d \mathop{\mathbf{E}}_{L_{i-1}, x^{(i)}} \left( (w_{L_{i-1}}^* \cdot x_i - w^* \cdot x_i)^2 \mid x^{(i)} \right) \\
&= \sum_{i=1}^d \mathop{\mathbf{E}}_{L_{i-1}, x^{(i)}} \left( (w_{L_{i-1}^{\perp}}^* \cdot x^{(i)})^2 \mid x^{(i)} \right) \\
&= \sum_{i=1}^d \mathop{\mathbf{E}}_{L_{i-1}, x^{(i)}} \left( \left( w_{L_{i-1}^{\perp}}^* \cdot (x^{(i)})_{L_{i-1}^{\perp}} \right)^2 \mid x^{(i)} \right) \quad (1) \\
&= \sum_{i=1}^d \mathop{\mathbf{E}}_{L_{i-1}, x^{(i)}} \left( \left\| w_{L_{i-1}^{\perp}}^* \right\|^2 \left\| (x_i)_{L_{i-1}^{\perp}} \right\|^2 (u \cdot a)^2 \right) \quad (2) \\
&= \sum_{i=1}^d \mathop{\mathbf{E}}_{L_{i-1}, x^{(i)}} \left\| w_{L_{i-1}^{\perp}}^* \right\|^2 \mathop{\mathbf{E}}_{L_{i-1}, x^{(i)}} \left\| (x^{(i)})_{L_{i-1}^{\perp}} \right\|^2 \mathop{\mathbf{E}}_{L_{i-1}, x^{(i)}} (u \cdot a)^2 \quad (3) \\
&= \sum_{i=1}^d \frac{d-i+1}{d} \frac{1}{d-i+1} \mathop{\mathbf{E}}_{L_{i-1}, x^{(i)}} \left\| (x^{(i)})_{L_{i-1}^{\perp}} \right\|^2 \\
&= \frac{1}{d} \sum_{i=1}^d \mathop{\mathbf{E}}_{L_{i-1}, x^{(i)}} \left\| (x^{(i)})_{L_{i-1}^{\perp}} \right\|^2 . \quad (4)
\end{aligned}
$$

Here in (1), we use the fact that $w_{L_{i-1}^{\perp}}^* \in L_{i-1}^{\perp}$. In (2), we denote by $u = w_{L_{i-1}^{\perp}}^* / \left\| w_{L_{i-1}^{\perp}}^* \right\|, a = (x^{(i)})_{L_{i-1}^{\perp}} / \left\| (x^{(i)})_{L_{i-1}^{\perp}} \right\|$. In (3), we use the fact that $(u \cdot a)^2$ is independent on the knowledge of $w_{L_{i-1}^{\perp}}^*$ and $\left\| (x^{(i)})_{L_{i-1}^{\perp}} \right\|$. Furthermore, we notice that $w_{L_{i-1}^{\perp}}^*$ is the projection of $w^*$ onto a random $d-i+1$ dimensional subspace. So we obtain $\mathbf{E} \left\| w_{L_{i-1}^{\perp}}^* \right\| = (d-i+1)/d$. Furthermore, we know

that given the knowlege of $w^*_{L^\perp_{i-1}}$, $a = (x^{(i)})_{L^\perp_{i-1}} / \left\| (x^{(i)})_{L^\perp_{i-1}} \right\|$ is uniformly drawn from the unit sphere in $L^\perp_{i-1}$. By Lemma B.1, we obtain $\mathbf{E}_{L_{i-1},x^{(i)}} (u \cdot a)^2 = 1/(d - i + 1)$. Notice that by Lemma B.1, $\mathbf{E}_{L_{i-1},x^{(i)}} \left\| (x^{(i)})_{L^\perp_{i-1}} \right\|^2 = (d - i + 1)/d$. This implies that $\mathbf{E}_{X,\mathcal{A}} L(\mathcal{A}, X, w^*) \leq 1/2$.

$\square$

## B.3 The Gain of Algorithm 1

In this section, we analyze the gain from ordering of Algorithm 1. Our main goal is to prove Proposition 4.1, which we restate as follows.

**Proposition B.4** (Proposition 4.1). *Let $X$ be a set of $n \geq \text{poly}(d)$ examples drawn i.i.d. from $\mathbb{S}^d$ and let $\mathcal{A}$ denote Algorithm 1, then over the randomness of the dataset $X$, in expectation, $\mathcal{G}(\mathcal{A}, X, \mathbb{S}^d)^{-1} \leq O(1/d + n^{-2/d})$.*

We know that $\min_{\mathcal{A}^{\text{random}}} \mathbf{E}_{X \sim D} \mathcal{L}(\mathcal{A}^{\text{random}}, X, \mathbb{S}^d) \in \Theta(1)$ from the last section. So, to analyze the gain of Algorithm 1, the central question is to upper bound $\mathcal{L}(\mathcal{A}, X, \mathbb{S}^d)$. To do this, we need the following probability lemma.

**Lemma B.5.** *Let $X_1, \ldots, X_m$ be $m$ independent random variable with distribution $\text{Beta}(k/2, (d - k)/2)$, $1 \leq k < d$. Denote by $Y = \min\{X_1, \ldots, X_m\}$. Then we have $\mathbf{E} Y \leq O((m/\log m)^{-2/k} k/d)$.*

*Proof of Lemma B.5.* We want to find some sufficient small $\epsilon > 0$ such that $\mathbf{E} Y \leq \epsilon$. Notice that $\mathbf{E} y \leq \mathbf{E} X_i \leq k/d$. So we can without loss of generality assume $\epsilon \leq k/d$. To get such an upper bound for $\mathbf{E} Y$, it is sufficient to find some $\epsilon > 0$ such that $\mathbf{Pr}(Y \geq \epsilon/2) \leq \epsilon/2$ because for such $\epsilon$, we have

$$\mathbf{E} Y \leq \mathbf{E}(Y \mid Y \leq \epsilon/2) + \mathbf{Pr}(Y \geq \epsilon/2) \leq \epsilon.$$

In the rest of the proof, we show $\epsilon = O((m/\log m)^{-2/k} k/d)$ satisfies such conditions by showing that such an $\epsilon$ satisfies some cleaner sufficient conditions for $\mathbf{Pr}(Y \geq \epsilon/2) \leq \epsilon/2$.

We use the fact that $\mathbf{Pr}(Y \geq y) = \mathbf{Pr}(X_i \geq y)^m = (1 - \mathbf{Pr}(X_i \leq y))^m \leq \exp(-m \mathbf{Pr}(X_i \leq y))$. If $\exp(-m \mathbf{Pr}(X_i \leq \epsilon/2)) \leq \epsilon/2$, then such $\epsilon$ satisfies $\mathbf{Pr}(Y \geq \epsilon/2) \leq \epsilon/2$. This is equivalent to find $\epsilon$ such that

$$m \mathbf{Pr}(X_i \leq \epsilon/2) \geq \log(2/\epsilon). \tag{5}$$

We next lower bound $\mathbf{Pr}(X_i \leq \epsilon/2)$. By Lemma B.1, We have

$$
\begin{aligned}
\mathbf{Pr}(X_i \leq \epsilon/2) &= B(\frac{k}{2}, \frac{d-k}{2})^{-1} \int_0^{\frac{\epsilon}{2}} x^{\frac{k}{2}-1}(1-x)^{\frac{d-k}{2}-1} dx \\
&\geq B(\frac{k}{2}, \frac{d-k}{2})^{-1} (1 - \epsilon/2)^{\frac{d-k}{2}} \int_0^{\frac{\epsilon}{2}} x^{\frac{k}{2}-1} dx \\
&\geq \frac{2}{k} B(\frac{k}{2}, \frac{d-k}{2})^{-1} \exp\left(-e\epsilon(d-k)/4\right) (\epsilon/2)^{k/2} \\
&\geq \frac{2}{k} B(\frac{k}{2}, \frac{d-k}{2})^{-1} \exp\left(-ek(d-k)/4d\right) (\epsilon/2)^{k/2} \\
&\geq \frac{2}{k} B(\frac{k}{2}, \frac{d-k}{2})^{-1} \exp\left(-ek/4\right) (\epsilon/2)^{k/2} \\
&= \frac{2}{k} \exp\left(-ek/4\right) (\epsilon/2)^{k/2} \frac{k(d-k)}{2d} \binom{d/2}{k/2} \\
&\geq \frac{d-k}{d} \exp\left(-ek/4\right) \left(\frac{d\epsilon}{2k}\right)^{k/2}.
\end{aligned}
$$

In the third inequality, we use the fact that $\epsilon < k/d$. To make (5) holds, we only need to find some $\epsilon$ such that

$$\frac{m(d-k)}{d} \exp\left(-ek/4\right) \left(\frac{d\epsilon}{2k}\right)^{k/2} \geq \log(2/\epsilon).$$

That can be done by choosing some $\epsilon = O((m/\log m)^{-2/k}k/d)$. $\qquad\square$

With Lemma B.5, we are able to give the following upper bound for $\mathcal{L}(\mathcal{A}, X, \mathbb{S}^d)$.

**Lemma B.6.** *Let $X$ be a set of $n \geq \mathrm{poly}(d)$ examples drawn i.i.d. from $\mathbb{S}^d$ and let $\mathcal{A}$ denote Algorithm 1, then $\mathbf{E}_X \mathcal{L}(\mathcal{A}, X, \mathbb{S}^d) \leq O(1/d + n^{-2/d})$.*

*Proof of Lemma B.6.* We know that with probability 1, $X$ is in general position. That is to say every $d$ examples in $X$ are linearly independent. So after seeing labels of $d$ different examples, Algorithm 1 learns $w^*$ exactly by solving a system of linear equations. Thus, we only need to bound the loss of Algorithm 1 on the first $d$ examples. Use a similar approach as we did in the proof of Lemma B.3, We have

$$
\mathbf{E}_{X,A} \mathcal{L}(\mathcal{A}, X, w^*) = \mathbf{E} \sum_{i=1}^{d} (y^{(i)} - \hat{y}^{(i)})^2 = \sum_{i=1}^{d} \mathbf{E}_{L_{i-1}, x^{(i)}} \left( (y^{(i)} - \hat{y}^{(i)})^2 \mid x^{(i)} \right)
$$

$$
= \sum_{i=1}^{d} \mathbf{E}_{L_{i-1}, x^{(i)}} \left( (w^*_{L_{i-1}} \cdot x_i - w^* \cdot x_i)^2 \mid x^{(i)} \right)
$$

$$
= \sum_{i=1}^{d} \mathbf{E}_{L_{i-1}, x^{(i)}} \left( (w^*_{L_{i-1}^\perp} \cdot x^{(i)})^2 \mid x^{(i)} \right) = \sum_{i=1}^{d} \mathbf{E}_{L_{i-1}, x^{(i)}} \left( \left( w^*_{L_{i-1}^\perp} \cdot (x^{(i)})_{L_{i-1}^\perp} \right)^2 \mid x^{(i)} \right)
$$

$$
= \sum_{i=1}^{d} \mathbf{E}_{L_{i-1}, x^{(i)}} \left( \left\| w^*_{L_{i-1}^\perp} \right\|^2 \left\| (x^{(i)})_{L_{i-1}^\perp} \right\|^2 (u \cdot a)^2 \right)
$$

$$
= \sum_{i=1}^{d} \mathbf{E}_{L_{i-1}, x^{(i)}} \left\| w^*_{L_{i-1}^\perp} \right\|^2 \mathbf{E}_{L_{i-1}, x^{(i)}} \left\| (x^{(i)})_{L_{i-1}^\perp} \right\|^2 \mathbf{E}_{L_{i-1}, x^{(i)}} (u \cdot a)^2
$$

$$
= \sum_{i=1}^{d} \frac{d-i+1}{d} \frac{1}{d-i+1} \mathbf{E}_{L_{i-1}, x^{(i)}} \left\| (x^{(i)})_{L_{i-1}^\perp} \right\|^2 = \frac{1}{d} \sum_{i=1}^{d} \mathbf{E}_{L_{i-1}, x^{(i)}} \left\| (x^{(i)})_{L_{i-1}^\perp} \right\|^2.
$$

To bound the loss of Algorithm 1, it remains to bound the $\mathbf{E}_{L_{i-1}, x^{(i)}} \left\| (x^{(i)})_{L_{i-1}^\perp} \right\|^2$, which is the minimum norm of the projection of $n/(d-i)^2$ random points onto a $d-i+1$ dimensional subspace for $i > 1$. By Lemma B.5, we know that $\mathbf{E}_{L_{i-1}, x^{(i)}} \left\| (x^{(i)})_{L_{i-1}^\perp} \right\|^2 \leq O((n/\log n)^{-2/(d-i)} \frac{d-i}{d}) \leq O((n/\log n)^{-2/d} \frac{d-i}{d})$. Sum these inequalities together, we obtain that $\mathbf{E}_{X,A} \mathcal{L}(\mathcal{A}, X, w^*) \leq O(1/d + (n/\log n)^{-2/d})$. We notice that if $n > d^d$, then $(n/\log n)^{-2/d} \leq 1/d$ and if $n \leq d^d$, $(\log n)^{2/d} = O(1)$. This implies that $\mathbf{E}_{X,A} \mathcal{L}(\mathcal{A}, X, w^*) \leq O(1/d + n^{-2/d})$.

$\qquad\square$

With Lemma B.6, we can obtain the proof of Proposition 4.1 immediately.

*Proof of Proposition 4.1.* By Lemma B.2, we know that $\min_{\mathcal{A}^{\mathrm{random}}} \mathbf{E}_{X \sim D} \mathcal{L}(\mathcal{A}^{\mathrm{random}}, X, \mathbb{S}^d) \in \Omega(1)$. By Lemma B.6, we know that $\mathbf{E}_X \mathcal{L}(\mathcal{A}, X, \mathbb{S}^d) = \mathbf{E}_{w^* \sim \mathbb{S}^d} \mathbf{E}_{X,A} \mathcal{L}(\mathcal{A}, X, w^*) \leq O(1/d + n^{-2/d})$. Thus, we have in expectation, $\mathcal{G}^{-1}(\mathcal{A}, X, \mathbb{S}^d) \leq O(1/d + n^{-2/d})$. $\qquad\square$

In particular, we remark that with a simple application of Markov inequality, Lemma B.6 actually implies $\mathcal{G}^{-1}(\mathcal{A}, X) \leq O(1/d + n^{-2/d})$, since Lemma B.6 holds for arbitrary $w^*$.

### B.4 The Gain of the Best Learner

In this subsection, we analyze the gain of the best learner and prove Proposition 4.2.

**Proposition B.7** (Proposition 4.2). *Let $X \subseteq S^d$ be a set of $n > \mathrm{poly}(d)$ examples drawn iid from $\mathbb{S}^d$. For every $\delta \in (3/n, 1)$, with probability at least 99%, $\mathcal{G}^*(X, \mathbb{S}^d)^{-1} \geq \Omega(1/d + \delta n^{-(2+2\delta)/d})$.*

By [Lemma B.3](), we know that $\min_{\mathcal{A}^{\text{random}}} \mathbf{E}_{X \sim D} \mathcal{L}(\mathcal{A}^{\text{random}}, X, \mathbb{S}^d) \leq O(1)$. So in the rest of this section, we will focus on the cost of the best learner over a random set of examples $X$. Our central result is the following information-theoretic lower bound.

**Lemma B.8.** *Let $X \subseteq S^d$ be a set of $n > \text{poly}(d)$ examples drawn iid from $\mathbb{S}^d$. For every $\delta \in (3/n, 1)$, with probability at least $99\%$, $X$ satisfies, for every self-directed learner $\mathcal{A}$, $\mathcal{L}(\mathcal{A}, X, \mathbb{S}^d) \geq \Omega(1/d + \delta n^{-(2+2\delta)/d})$.*

*Proof of [Proposition 4.2]().* According to [Lemma A.3](), the best learner $\mathcal{A}^*$ to learn $w^* \sim \mathbb{S}^d$ is the one that selects a sequence of examples with minimum spanning cost and makes the Bayesian optimal prediction. Let $\sigma^*$ be the sequence of examples of $X$ with the minimum spanning cost. Then we know that $\mathcal{L}(\mathcal{A}^*, X, \mathbb{S}^d) = C(\sigma^*, m)/d$. So in the rest of the proof, we will show that with $99\%$, there is no sequence of examples with a very small spanning cost. We will first show that for a fixed permutation of the $n$ random examples, the probability that the sequence has a very small spanning cost is small, then we will apply a union bound to finish the proof.

We consider the random variable $Y$ defined as follows. Let $V$ be a fixed subspace with dimension $k \geq d/2$, and let $x$ be a uniform vector over $S^{d-1}$. Denote by $Y = \|x_V\|^2$. By [Lemma B.1](), we know that $Y \sim \text{Beta}(k/2, (d-k)/2)$. We show that with a very high probability $Y$ is large. We have for every $\epsilon \in (0, 1)$,

$$
\begin{aligned}
\mathbf{Pr}\,(Y \leq \epsilon) &= B(\frac{k}{2}, \frac{d-k}{2})^{-1} \int_0^\epsilon x^{\frac{k}{2}-1}(1-x)^{\frac{d-k}{2}-1} dx \\
&\leq \int_0^\epsilon x^{\frac{k}{2}-1}(1-x)^{\frac{d-k}{2}-1} dx \frac{k(d-k)}{2d} \left(\frac{ed}{k}\right)^{k/2} \\
&\leq (ed/k)^{k/2}(k/2d)\epsilon^{k/2} \leq (ed/k)^{k/2}\epsilon^{k/2}.
\end{aligned}
$$

We set up $\epsilon = \frac{k}{2ed} n^{-(2+2\delta)/k}$, then we obtain that with probability at most $n^{-(1+\delta)}/2$,

$$
Y \leq \epsilon \leq \frac{k}{ed} n^{-(2+2\delta)/k} \leq n^{-(2+2\delta)/k}/2.
$$

Now, we fix a permutation $\sigma$ of the $n$ random examples. Without loss of generality, we assume this sequence is $x^{(1)}, \ldots, x^{(n)}$. We will show a slightly stronger statement that $C(\sigma, d/2)$ is large with an extremely high probability.

Now assume that the spanning cost of this sequence $x^{(1)}, \ldots, x^{(d/2)}$ is small, say, $\sum_{i=2}^{d/2} \left\| (x^{(i)})_{L_{i-1}^\perp} \right\|^2 \leq \delta d n^{-(2+2\delta)/k}/12$. This implies that at most $\delta d/6$ terms of the first $d/2$ terms can be larger than $n^{-(2+2\delta)/d}/2$. Notice that since $x^{(1)}, \ldots, x^{(d/2)}$ are independent for a fixed permutation, we obtain that

$$
\begin{aligned}
\mathbf{Pr}\left(\sum_{i=2}^{d/2} \left\| (x^{(i)})_{L_{i-1}^\perp} \right\|^2 \leq \delta d n^{-(2+2\delta)/k}/12\right) &\leq (\frac{1}{2})^{(1-\delta/6)d} n^{-\frac{(1+\delta)(1-\delta/6)d}{2}} \binom{d/2}{\delta d/6} \\
&\leq 0.99 n^{-\frac{(1+\delta)(1-\delta/6)d}{2}} (3/\delta)^{\delta d/6} \\
&\leq 0.99 n^{-\frac{(1+\delta)(1-\delta/6)d}{2} + \frac{\delta d}{6}} \leq 0.99 n^{-d/2}.
\end{aligned}
$$

In the third inequality, we use the fact that $3/\delta \leq n$, and in the last inequality we use the fact that $\delta < 1$. Since there are at most $n^{d/2}$ such permutations for the first $d/2$ terms, by union bound, we obtain that with probability $0.99$ every sequence of examples has a spanning cost at least $\Omega(1 + \delta d n^{-(2+2\delta)/d})$, where the constant term comes from the spanning cost of the first term. This implies that for every self-directed learner $\mathcal{A}$, $\mathcal{L}(\mathcal{A}, X, \mathbb{S}^d) \geq \Omega(1/d + \delta n^{-(2+2\delta)/d})$. $\qquad\square$

With [Lemma B.8](), we are able to obtain the proof of [Proposition 4.2]() immediately.

*Proof.* By Lemma B.3, we know that $\min_{\mathcal{A}^{\mathrm{random}}} \mathbf{E}_{X \sim D} \mathcal{L}(\mathcal{A}^{\mathrm{random}}, X, \mathbb{S}^d) \leq O(1)$ and by Lemma B.8, we know that with probability at least 90%, for every learner $\mathcal{A}$, $\mathcal{L}(\mathcal{A}, X, \mathbb{S}^d) \geq \Omega(1/d + \delta n^{-(2+2\delta)/d})$. This implies that $\mathcal{G}^*(X, \mathbb{S}^d)^{-1} \geq \Omega(1/d + \delta n^{-(2+2\delta)/d})$ with probability at least 99%. $\square$

We remark that a direct implication of Lemma B.8 is that for every learner $\mathcal{A}$, there is some $w^*$ such that with probability at least 99%, the learning cost $L(\mathcal{A}, X, w^*)$ is at least $\Omega(1/d + \delta n^{-(2+2\delta)/d})$, which means $\mathcal{G}^*(X) \geq \Omega(1/d + \delta n^{-(2+2\delta)/d})$.

# C    Missing Proofs from Section 5

In this section, we analyze the performance of Algorithm 2 for the self-directed ReLU regression problem and present the proof of Theorem 5.1 in this section.

**Theorem C.1.** *Let $X \subseteq \mathbb{R}^d$ be a set of $n \geq \mathrm{poly}(d)$ examples drawn i.i.d. from $\mathbb{S}^d$. Let $\mathcal{G}(\mathcal{A}, X)$ be the gain from ordering of Algorithm 2 for the self-directed ReLU regression problem over $X$. Then,*

$$\mathcal{G}(\mathcal{A}, X)^{-1} \leq O(\frac{1}{d}) + \begin{cases} \min\{O(\frac{\tan^2 \theta_0 \log d}{d}), O(\frac{\log d}{n^{2/d}})\} & \text{if } \frac{n\theta_0}{4\pi d \log d} > \exp(\frac{d}{8}), \\ \min\{O(\frac{\tan^2 \theta_0 \log d}{\log(n\theta_0)}), O(\frac{\log d}{n^{2/d}})\} & \text{if } 1 \leq \frac{n\theta_0}{4\pi d \log d} \leq \exp(\frac{d}{8}) \\ O(\frac{1}{n}) & \text{if } \frac{n\theta_0}{4\pi d \log d} < 1. \end{cases}$$

## C.1    ReLU Regression with A Random Order

We start with an information-theoretic lower bound for the cost of a learner that uses a random order.

**Lemma C.2.** *Let $w^* \in \mathbb{R}^d$ be a target vector drawn uniformly from $\mathbb{S}^d$. Let $X$ be a set of $n > d$ examples drawn i.i.d. from $\mathbb{S}^d$. Let $\mathcal{A}$ be any learning algorithm for the ReLU regression problem over $X$ that uses a random order to learn $w^*$. For a random set of examples, let the random variable $\mathcal{L}(\mathcal{A}, X, \mathbb{S}^d)$ be the expected learning cost of $\mathcal{A}$. Then we have $\mathbf{E}_X \mathcal{L}(\mathcal{A}, X, \mathbb{S}^d) \geq \Omega(1)$.*

*Proof of Lemma C.2.* Assume a self-directed learner $\mathcal{A}$ learns $w^*$ with a random sequence of examples $x^{(1)}, \ldots, x^{(n)}$. To lower bound the learning cost of $\mathcal{A}$, we slightly relax the learning model so that in each round $\mathcal{A}$ not only sees $\mathrm{Relu}(w^* \cdot x^{(i)})$ but also sees $w^* \cdot x^{(i)}$. In the relaxed learning model, $w^*_{L_{i-1}}$ is known at the beginning of each round. We say an example $x^{(i)}$ is good if $w^*_{L_{i-1}} \cdot x^{(i)} \geq 0$ and $w^*_{L^\perp_{i-1}} \cdot x^{(i)} \geq 0$. If $x^{(i)}$ is not good, then we say it is a bad example. Notice that by symmetry, in each round with probability $1/4$, we will get a good example. Let $y^{(i)}$ be the true label of $x^{(i)}$ and let $\hat{y}^{(i)}$ be the prediction for $x^{(i)}$. Consider a single learning round, we have

$$\mathbf{E}(y^{(i)} - \hat{y}^{(i)})^2 = \mathbf{Pr}(x^{(i)} \text{ is good}) \mathbf{E}\left((y^{(i)} - \hat{y}^{(i)})^2 \mid x^{(i)} \text{ is good}\right) + \mathbf{Pr}(x^{(i)} \text{ is bad}) \mathbf{E}\left((y^{(i)} - \hat{y}^{(i)})^2 \mid x^{(i)} \text{ is bad}\right)$$

$$\geq \frac{1}{4} \mathbf{E}\left((y^{(i)} - \hat{y}^{(i)})^2 \mid x^{(i)} \text{ is good}\right)$$

$$= \frac{1}{4} \mathop{\mathbf{E}}_{L_{i-1}, a_{i-1}, x^{(i)}} \left((\hat{y}_i - a_{i-1} \cdot x^{(i)} - w^*_{L^\perp_{i-1}} \cdot x^{(i)})^2 \mid x^{(i)} \text{ is good}, w^*_{L_{i-1}} = a_{i-1}, x^{(i)}\right)$$

$$= \frac{1}{4} \mathop{\mathbf{E}}_{L_{i-1}, a_{i-1}, x_i} \left((\hat{p}_i - w^*_{L^\perp_{i-1}} \cdot x^{(i)})^2 \mid x^{(i)} \text{ is good}, w^*_{L_{i-1}} = a_{i-1}, x^{(i)}\right)$$

$$\geq \frac{1}{4} \mathbf{Var}\left((w^*_{L^\perp_{i-1}} \cdot x^{(i)}) \mid x^{(i)} \text{ is good}, w^*_{L_{i-1}} = a_{i-1}, x^{(i)}\right)$$

$$= \frac{1}{4} \mathbf{Var}\left((w^*_{L^\perp_{i-1}} \cdot (x^{(i)})_{L^\perp_{i-1}}) \mid x^{(i)} \text{ is good}, w^*_{L_{i-1}} = a_{i-1}, x^{(i)}\right)$$

$$= \frac{1}{4} \frac{d - i + 1}{d} \frac{d - i + 1}{d} \mathbf{Var}\left((\frac{w^*_{L^\perp_{i-1}}}{\left\|w^*_{L^\perp_{i-1}}\right\|} \cdot \frac{(x^{(i)})_{L^\perp_{i-1}}}{\left\|(x^{(i)})_{L^\perp_{i-1}}\right\|}) \mid x^{(i)} \text{ is good}, w^*_{L_{i-1}} = a_{i-1}, x^{(i)}\right)$$

$$\geq \Omega(\frac{d - i + 1}{d^2}).$$

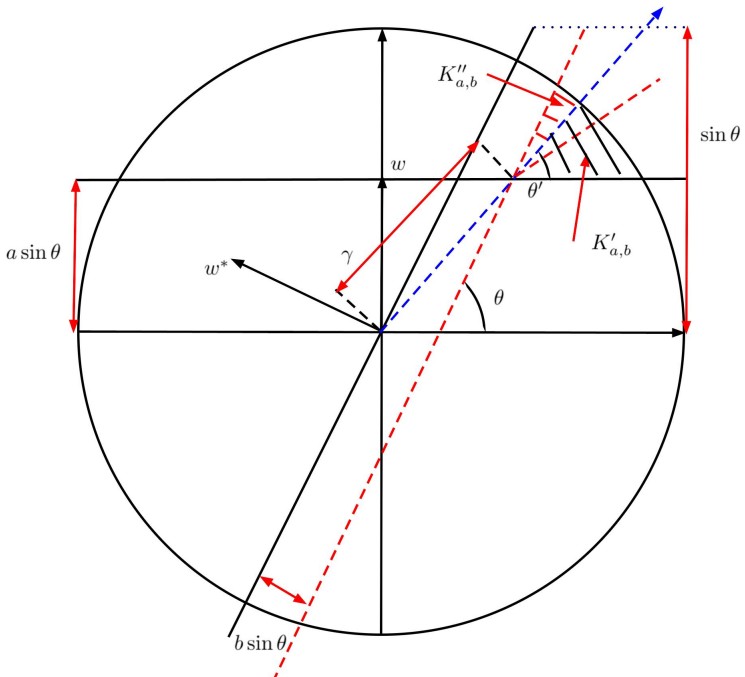

Figure 1: Geometry illustration for Lemma 5.2. $K''_{a,b}$ is the red shadowed area and $K'_{a,b}$ is the black shadowed area. As $a/\gamma$ increases, $\theta' = \arcsin(a\sin\theta/\gamma) > \theta/2$ and $M(K''_{a,b}) \subseteq K'_{a,b}$.

Here in the third equality, we denote by $\hat{p}_i = \hat{y}^{(i)} - a_{i-1} \cdot x^{(i)}$. This implies that $\mathbf{E}_X \, \mathcal{L}(\mathcal{A}, X, \mathbb{S}^d) = \sum_{i=1}^d \mathbf{E}(y^{(i)} - \hat{y}^{(i)})^2 = \Omega(1)$.

$\square$

Lemma C.2 implies that for every $\mathcal{A}^{\mathrm{random}}$, there is some $w^*$ such that $\mathbf{E}_X \, L(\mathcal{A}, X, w^*) = \Omega(1)$. Thus, $\min_{\mathcal{A}^{\mathrm{random}}} \max_{w^*} \mathbf{E}_X \, L(\mathcal{A}^{\mathrm{random}}, X, w^*) \in \Omega(1)$.

## C.2   Geometric Technical Lemma

As Lemma C.2 suggests, to analyze the gain of Algorithm 2, we only need to analyze the learning cost of Algorithm 2. The central technique we use is the following geometric technical lemma. At a high lever, it bounds by above the probability that a point far away from some guess $w$ is also far away from the ground-truth vector $w^*$. We define by $K_a := \{x \in C \mid w \cdot x \geq a\sin\theta\}$ the set of points that are far away from the guess $w$ (large-margin points) and by $K_{a,b} := \{x \in C \mid w \cdot x \geq a\sin\theta, w^*x \leq -b\sin\theta\}$ the subset of $K_a$ that contains points that are far from $w^*$ (large-margin with respect to $w^*$). We show that conditional on $K_a$ the probability that $x \in K_{a,b}$ decays exponentially fast as a function of the margin $b$, i.e., $\mathbf{Pr}\left(x \in K_{a,b} \mid x \in K_a\right) \leq \exp\left(-O(db^2 + 2ab\cos\theta)\right)$ This means that with very high probability all points in $K_a$ will land "close" to the target-vector $w^*$.

**Lemma C.3** (Lemma 5.2). *Let $w^* \in S^d$ be a target vector and let $w \in S^d$ be an arbitrary vector such that $\theta = \theta(w^*, w) < \pi/2$. Denote by $C = \{x \in S^d \mid w^*x \leq 0, w \cdot x \geq 0\}$. For every $a, b \in (0,1)$, denote by $K_a := \{x \in C \mid w \cdot x \geq a\sin\theta\}$ and $K_{a,b} := \{x \in C \mid w \cdot x \geq a\sin\theta, w^*x \leq -b\sin\theta\}$. Let $x$ be a point uniformly drawn from $S^d$. There is some absolute constant $c > 1$ such that if $a/b \geq c$ then $\mathbf{Pr}\left(x \in K_{a,b} \mid x \in K_a\right) \leq 2\exp\left(-\frac{d}{3(1-a^2)}(b^2 + 2ab\cos\theta)\right)$.*

*Proof of Lemma 5.2.* Without loss of generality, we assume that $w = e_2$ and $w^* = -\sin\theta e_1 + \cos\theta e_2$, where $e_1, e_2$ are the first two standard basis of $\mathbb{R}^d$. We consider the projection of $x$ onto the 2-dimensional subspace spanned by $w, w^*$. We use a polar coordinate to write such a projection as

$r\cos\phi e_1 + r\sin\phi e_2$. In this way, we obtain that

$$C = \{(r,\phi) \mid r \in (0,1), \phi \in (0,\phi)\},$$
$$K_a = \{(r,\phi) \mid r\sin\phi \geq a\sin\theta, r \in (0,1), \phi \in (0,\phi)\},$$
$$K_{a,b} = \{(r,\phi) \mid r\sin(\phi-\theta) \geq -b\sin\theta, r \in (0,1), \phi \in (0,\phi)\}.$$

We partition $K_{a,b}$ into two parts. (See Figure 1 for detail.) Let $\gamma^2 = a^2 + b^2 + 2ab\cos\theta$ Denote by $K'_{a,b} := \{(r,\phi) \in K_{a,b} \mid \sin\phi \in [a\sin\theta, a\sin\theta/\gamma]\}$ and $K''_{a,b} = \{(r,\phi) \in K_{a,b} \mid \sin\phi \in [a\sin\theta/\gamma, (\sqrt{1-(b\sin\theta)^2} - b\cos\theta)\sin\theta]\}$. We mirror $K''_{a,b}$ according to ray $\{(r,\phi) \mid \sin\phi = a\sin\theta/\gamma\}$ and obatin a mirror set $M(K''_{a,b})$. We notice that when $a/b > c$ for some constant $c > 0$, $a/\gamma > 1/\sqrt{2}$ and $\sin\theta' = a\sin\theta/\gamma \geq \sin(\theta/2)$. This implies $M(K''_{a,b}) \subseteq K'_{a,b}$. By symmetry, we obtain that $\mathbf{Pr}(x \in K''_{a,b}) \leq \mathbf{Pr}(x \in K'_{a,b})$ when this happens. Since $K_{a,b} \subseteq K_a$, we have

$$\mathbf{Pr}\left(x \in K_{a,b} \mid x \in K_a\right) \leq 2\mathbf{Pr}\left(x \in K'_{a,b} \mid x \in K_a\right).$$

In the rest of the proof, we will show $\mathbf{Pr}\left(x \in K'_{a,b} \mid x \in K_a\right) \leq \exp\left(-\frac{d}{2(1-a^2)}(b^2 + 2ab\cos\theta)\right)$

By Lemma B.1, recall the density function of $r$ is $f(r) = \frac{d-2}{2\pi}(1-r^2)^{d/2-2}r$. Furthermore, $\phi \sim U[0,2\pi]$. This implies that

$$\mathbf{Pr}\left(x \in K_a\right) = \frac{d-2}{(2\pi)^2}\int_{\arcsin(a\sin\theta)}^{\theta}\int_{\frac{a\sin\theta}{\sin\phi}}^{1}(1-r^2)^{d/2-2}r\,dr\,d\phi$$
$$= \frac{1}{(2\pi)^2}\int_{\arcsin(a\sin\theta)}^{\theta}(1-(\frac{a\sin\theta}{\sin\phi})^2)^{d/2-1}\,d\phi.$$
$$\mathbf{Pr}\left(x \in K'_{a,b}\right) = \frac{d-2}{(2\pi)^2}\int_{\arcsin(a\sin\theta)}^{\arcsin(a\sin\theta/\gamma)}\int_{\frac{a\sin\theta}{\sin\phi}}^{1}(1-r^2)^{d/2-2}r\,dr\,d\phi$$
$$= \frac{1}{(2\pi)^2}\int_{\arcsin(a\sin\theta)}^{\arcsin(a\sin\theta/\gamma)}(1-(\frac{a\sin\theta}{\sin\phi})^2)^{d/2-1}\,d\phi.$$

To simplify the notation, we introduce the following function

$$g(t) = \frac{1}{(2\pi)^2}\int_{\arcsin(a\sin\theta)}^{\arcsin(a\sin\theta/t)}(1-(\frac{a\sin\theta}{\sin\phi})^2)^{d/2-1}\,d\phi, \quad t \in [a,1].$$

Then we have We are interested in the upper bound for $g(\gamma)/g(a)$. To do this, we upper bound

$$\log\left(g(\gamma)/g(a)\right) = \left(\log g(t)\right)'\left(\gamma - a\right).$$

for some $t \in [a,\gamma]$. Now we derive an upper bound for $\log g(t)$. Notice that

$$g'(t) = -\frac{1}{(2\pi)^2}\frac{a\sin\theta(1-t^2)^{d/2-1}}{t^2\sqrt{1-(\frac{a\sin\theta}{t})^2}} = -\frac{1}{(2\pi)^2}\frac{a\sin\theta(1-t^2)^{d/2-1}}{t\sqrt{t^2-(a\sin\theta)^2}}.$$

On the other hand, we have $\mathbf{Pr}\left(x \in K'_{a,b} \mid x \in K_a\right) = g(\gamma)/g(a)$.

$$g(t) = \frac{1}{(2\pi)^2}\int_{\arcsin(a\sin\theta)}^{\arcsin(a\sin\theta/t)}(1-(\frac{a\sin\theta}{\sin\phi})^2)^{d/2-1}\,d\phi$$
$$= \frac{1}{(2\pi)^2}\int_{t}^{1}(1-s^2)^{d/2-1}\frac{a\sin\theta}{s\sqrt{s^2-(a\sin\theta)^2}}\,ds$$
$$\leq \frac{1}{(2\pi)^2}\int_{t}^{1}s(1-s^2)^{d/2-1}\,ds\frac{a\sin\theta}{t^2\sqrt{t^2-(a\sin\theta)^2}}$$
$$= \frac{(1-t^2)^{d/2}a\sin\theta}{(2\pi)^2dt^2\sqrt{t^2-(a\sin\theta)^2}}.$$

We have

$$(\log g(t))' = \frac{g'(t)}{g(t)} \le -dt(1-t^2)^{-1} \le -da(1-a^2)^{-1}.$$

So, we obtain that

$$\log\left(\frac{g(\gamma)}{g(a)}\right) \le -da(1-a^2)^{-1}(\gamma-a) = -\frac{da(\gamma^2-a^2)}{(\gamma+a)} \le -\frac{d(\gamma^2-a^2)}{3} = -\frac{d(b^2+2ab\cos\theta)}{3}.$$

Here, the second inequality holds because $\gamma = \sqrt{a^2 + b^2 + 2ab\cos\theta} \le 2a$. Thus, we get $\mathbf{Pr}\left(x \in K'_{a,b} \mid x \in K_a\right) \le \exp(-\frac{d(\gamma^2-a^2)}{3}) = \exp(-\frac{d(b^2+2ab\cos\theta)}{3})$. Put everything together, we obtain that $\mathbf{Pr}\left(x \in K_{a,b} \mid x \in K_a\right) \le 2\exp\left(-\frac{d}{3(1-a^2)}(b^2+2ab\cos\theta)\right).$

$\square$

We briefly explain how we will make use of this lemma. In Lemma 5.2, $w^*$ can be understood as the target vector and $w$ can be understood as our current hypothesis $w^{(i)}$ used in Algorithm 2. $\mathbf{Pr}\left(x \in K_{a,b} \mid x \in K_a\right)$ can be understood as if the margin $w^{(i)} \cdot x$ is large, then with high probability, the learner's cost $(w^* \cdot x)^2$ must be small.

Next, we will make use of the "maximum margin" lemma used in [DKTZ23] to show that the example we selected actually has a large margin.

**Lemma C.4** (Maximum Margin (Proposition 21 in [DKTZ23])). *Let $v, u \in \mathbb{R}^d$ be unit vectors such with angle $\theta(u,v) = \theta$. Let $C = \{x \in S^d \mid v \cdot x \le 0, u \cdot x \ge 0\}$. Let $x^{(1)}, \dots, x^{(n)}$ be $n$ i.i.d. examples drawn uniformly from $S^d$. Then*

1. *For all $n, s \ge 1$ and $c \ge 2$ such that $\exp(-dc/4) \le 4\pi s/(n\theta) \le 1$,*

$$\mathbf{Pr}\left(\max_{i\in[n]}(u \cdot x\mathbf{1}_{\{x^{(i)}\in C\}}) \le \sqrt{\frac{\log((n\theta/4\pi s))}{2cd}}\sin\theta\right) \le 2\exp(-2s).$$

2. *For all $n, s \ge 1$ such that $4\pi s/n\theta \le 1$,*

$$\mathbf{Pr}\left(\max_{i\in[n]}(u \cdot x\mathbf{1}_{\{x^{(i)}\in C\}}) \le \left(1-(\frac{4\pi s}{n\theta})^{2/d}\right)\sin\theta\right) \le 2\exp(-2s).$$

### C.3 Proof of Theorem 5.1

Now we are able to prove Theorem 5.1.

*Proof of Theorem 5.1.* We notice that if $(x^{(0)}, y^{(0)}) = (0,0)$, then with probability at least 99%, the first time we see a positive example, we will select some example $x^{(1)} \in S^d$ such that $0 \le w^* \cdot x^{(1)} \le O(1/d)$. This implies that we will pay $O(1/d)$ to get a pair of example $(x^{(1)}, y^{(1)})$ such that $\theta(x^{(1)}, w^*) \le \theta_0 = \pi/2$. So it sufficient to deal with the case where $(x^{(0)}, y^{(0)}) \ne (0,0)$. In this case, for every $i \in [d]$, we use random variable $c_i$ to denote the cost of Algorithm 2 for predicting labels of examples in $X_i$. Let $\theta_i = \theta(w^{(i)}, w^*)$ and $C_i := \{x \in X_i \mid w^{(i)} \cdot x \le 0, w^* \cdot x \ge 0\}$. We say Algorithm 2 makes a mistake at some example $x$ if $\mathrm{Relu}(w^{(i)} \cdot x) \ne \mathrm{Relu}(w^* \cdot x)$. We will derive a high probability bound for each $c_i$, then the total learning cost is $\sum_{i=1}^{d} c_i$ since after seeing $d$ positive examples, we are able to solve a linear equation to get $w^*$ exactly. We consider three cases here.

**Case 1:** $\theta_i \le \frac{4\pi d\log d}{n}$. In this case, our current hypothesis $w^{(i)}$ is very close to the target $w^*$. In this case, we have to consider two different situations. In the first situation, there is at least one example $x \in C_i$. Since Algorithm 2 keeps selecting example $x$ that minimizes $w^{(i)} \cdot x$, once we make a mistake, we must make a mistake at some example $x \in C_i$ by paying $(w^* \cdot x)^2 \le \sin^2\theta_i$. In the second situation, there is no example $x \in C_i$. Let $C'_i = \{x \in S^d \mid -x \in C_i\}$. By symmetry, we know that with probability at least $1 - 1/\mathrm{poly}(d)$, there are at most $O(\log d)$ examples in $C'_i$.

Since there is no example in $C_i$, Algorithm 2 will make a mistake at examples $x$ such that $w^* \cdot x \geq 0$ and will stop making mistakes at some example $x$ such that $w^* \cdot x > 0$. When we make a mistake at an example $x \in C_i'$, we pay $(w^{(i)} \cdot x)^2 \leq \sin^2 \theta_i$. Since there are at most $O(\log d)$ examples in $C_i'$, we will pay at most $O(\log d \sin^2 \theta_i)$ for examples in $C_i'$. When we make a mistake at a positive example $x$, we pay $(w^{(i)} \cdot x - w^* \cdot x)^2 = (w^*_{L_{i-1}^\perp} \cdot x)^2 \leq \sin^2 \theta_i$. In summary, with probability at least $1 - 1/\mathrm{poly}(d)$, we have $c_i \leq O(\log d) \sin^2 \theta_i \leq O(\frac{d^2 \log^3 d}{n^3}) \leq 1/(dn)$ since we assume that $n \geq \mathrm{poly}(d)$.

**Case 2:** $1 \leq \frac{n\theta_i}{4\pi d \log d} \leq \exp(\frac{d}{8})$. Since each example is drawn i.i.d. from $\mathbb{S}^d$ and $X_i$ is a random subset of them, we obtained that with probability $1 - 1/\mathrm{poly}(d)$, there must be at least one example $x \in C_i$ and Algorithm 2 will make its mistake at some example $x \in C_i$. Since Algorithm 2 always selects the example $x$ that minimize $w^{(i)} \cdot x$, we must make a mistake for an example $x \in \mathrm{argmin}_{x' \in C_i} w^{(i)} \cdot x'$ and pay $(w^* \cdot x)^2$. Now let $x \in \mathrm{argmin}_{x' \in C_i} w^{(i)} \cdot x'$ be the example where Algorithm 2 makes a mistake. Assume $w^{(i)} \cdot x = -a \sin \theta_i$ and $w^* \cdot x = b \sin \theta_i$, then our cost in this round is $b^2 \sin^2 \theta_i$. Denote by $K_a := \{x \in C_i \mid w^{(i)} \cdot x \leq -a \sin \theta_i\}$ and $K_{a,b} = \{x \in C_i \mid w^{(i)} \cdot x \leq -a \sin \theta_i, w^* \cdot x \geq b \sin \theta_i\}$. We want to show that given $w^{(i)} \cdot x$ is negative enough, with high probability, $b$ cannot be too large. By Lemma 5.2, this is

$$\mathbf{Pr}\left(x \in K_{a,b} \mid x \in K_a\right) \leq 2\exp\left(-\frac{d}{3(1-a^2)}(b^2 + 2ab\cos\theta)\right). \tag{6}$$

To derive a high probability bound for $b$, it remains to understand how large $a$ could be. Denote by $m = n/d$, which is the number of examples in $X_i$. Since $1 \leq \frac{m\theta_i}{4\pi \log d} \leq \exp(\frac{d}{8})$, by Lemma C.4, we obtain that with probability at least $1 - 1/\mathrm{poly}(d)$,

$$w^{(i)} \cdot x \leq -\sqrt{\frac{\log((m\theta_i/4\pi \log d))}{4d}} \sin \theta_i.$$

Since $1 \leq \frac{m\theta_i}{4\pi \log d}$, again, by Lemma C.4, we obtain that with probability at least $1 - 1/\mathrm{poly}(d)$, we have

$$w^{(i)} \cdot x \leq -\left(1 - (\frac{4\pi \log d}{m\theta_i})^{2/d}\right) \sin \theta_i.$$

This is to say, with probability at least $1 - 1/\mathrm{poly}(d)$, we have $w^{(i)} \cdot x \leq \min\{-\sqrt{\frac{\log((m\theta_i/4\pi \log d))}{4d}} \sin \theta_i, -\left(1 - (\frac{4\pi \log d}{m\theta_i})^{2/d}\right) \sin \theta_i\}$. Now we are able to derive the bound of $c_i$. Apply (6) with $a = \sqrt{\frac{\log((m\theta_i/4\pi \log d))}{4d}}$ and $b = \sqrt{\frac{16 \log^2 d}{d \cos^2 \theta_i \log(m\theta_i/4\pi \log d)}}$ we have

$$\mathbf{Pr}\left(x \in K_{a,b} \mid x \in K_a\right) \leq 2\exp(-\frac{2dab\cos\theta_i}{3}) \leq 1/\mathrm{poly}(d).$$

On the other hand, apply (6) with $a = 1 - (\frac{4\pi \log d}{m\theta_i})^{2/d}$ and $b = \sqrt{\frac{\log d}{d}(\frac{4\pi \log d}{m\theta_i})^{2/d}}$ we have

$$\mathbf{Pr}\left(x \in K_{a,b} \mid x \in K_a\right) \leq 2\exp(-\frac{2db^2}{3(1-a^2)}) \leq 1/\mathrm{poly}(d).$$

Combine these two bounds together, we obtain that with probability at least $1 - 1/\mathrm{poly}(d)$, $c_i \leq \min\{\frac{16 \log^2 d}{d \cos^2 \theta_i \log(m\theta_i/4\pi \log d)} \sin^2 \theta_i\}, \frac{\log d}{d}(\frac{4\pi \log d}{m})^{2/d}\} = \min\{O(\frac{\tan^2 \theta_i \log d}{d \log(n\theta_i)}), O(\frac{\log d}{dn^{2/d}})\}$.

**Case 3:** $\frac{n\theta_i}{4\pi d \log d} > \exp(\frac{d}{8})$. This case is similar to Case 2. We know that with probability $1 - 1/\mathrm{poly}(d)$ there must be at least 1 example in $C_i$ and we will make a mistake at that example. Since $1 \leq \frac{m\theta_i}{4\pi \log d}$, by Lemma C.4, we have $w^{(i)} \cdot x \leq -\left(1 - (\frac{4\pi \log d}{m\theta_i})^{2/d}\right) \sin \theta_i$. The slightly tricky part is that since $\frac{m\theta_i}{4\pi \log d} > \exp(\frac{d}{8})$, we are not able to apply the first case of Lemma C.4. However, if we sample $m'$ examples from $X_i$ such that $\frac{m\theta_i}{4\pi \log d} = \exp(\frac{d}{8})$ and consider the examples among the sampled points in $C_i$, then we are able to apply the first case of Lemma C.4. We will get

with probability at least $1 - 1/\text{poly}(d)$, $w^{(i)} \cdot x \le -\sqrt{\frac{1}{32}} \sin \theta_i$. Now apply Lemma 5.2, we know that with probability at least $1 - 1/\text{poly}(d)$, $c_i \le \min\{O(\frac{\tan^2 \theta_i \log d}{d^2}), O(\frac{\log d}{dn^{2/d}})\}$. In summary, if we define

$$
f(\theta) = \begin{cases} \min\{O(\frac{\tan^2 \theta \log d}{d^2}), O(\frac{\log d}{dn^{2/d}})\} & \text{if } \frac{n\theta}{4\pi d \log d} > \exp(\frac{d}{8}), \\ \min\{O(\frac{\tan^2 \theta \log d}{d \log(n\theta)}), O(\frac{\log d}{dn^{2/d}})\} & \text{if } 1 \le \frac{n\theta}{4\pi d \log d} \le \exp(\frac{d}{8}) \\ O(\frac{1}{dn}) & \text{if } \frac{n\theta}{4\pi d \log d} < 1, \end{cases}
$$

then for each $i \in [d]$, with probability at least $1 - 1/\text{poly}(d)$, $c_i \le f(\theta_i)$. We notice that $\cos \theta_i = \frac{w^{(i)} \cdot w^*}{\|w^{(i)}\| \|w^*\|} = \frac{\|w^*_{L_{i-1}}\|}{\|w^*\|}$, which implies that $\theta_i$ is decreasing in each round. Also, it is not hard to check as $\theta_i$ decreasing the cost bound $f(\theta_i)$ is also decreasing in each round. This implies that with probability at least $1 - 1/\text{poly}(d)$, we have $\sum_{i=1}^d c_i \le df(\theta_0)$. Combine with the cost for initialization, we have

$$
\mathcal{L}(\mathcal{A}, X) \le O(\frac{1}{d}) + \begin{cases} \min\{O(\frac{\tan^2 \theta_0 \log d}{d}), O(\frac{\log d}{n^{2/d}})\} & \text{if } \frac{n\theta_0}{4\pi d \log d} > \exp(\frac{d}{8}), \\ \min\{O(\frac{\tan^2 \theta_0 \log d}{\log(n\theta_0)}), O(\frac{\log d}{n^{2/d}})\} & \text{if } 1 \le \frac{n\theta_0}{4\pi d \log d} \le \exp(\frac{d}{8}) \\ O(\frac{1}{n}) & \text{if } \frac{n\theta_0}{4\pi d \log d} < 1. \end{cases}
$$

By Lemma C.2, we know that $\min_{\mathcal{A}^{\text{random}}} \max_{w^*} \mathbf{E}_X L(\mathcal{A}^{\text{random}}, X, w^*) \in \Omega(1)$. Thus, we have

$$
\mathcal{G}(\mathcal{A}, X)^{-1} \le O(\frac{1}{d}) + \begin{cases} \min\{O(\frac{\tan^2 \theta_0 \log d}{d}), O(\frac{\log d}{n^{2/d}})\} & \text{if } \frac{n\theta_0}{4\pi d \log d} > \exp(\frac{d}{8}), \\ \min\{O(\frac{\tan^2 \theta_0 \log d}{\log(n\theta_0)}), O(\frac{\log d}{n^{2/d}})\} & \text{if } 1 \le \frac{n\theta_0}{4\pi d \log d} \le \exp(\frac{d}{8}) \\ O(\frac{1}{n}) & \text{if } \frac{n\theta_0}{4\pi d \log d} < 1. \end{cases}
$$

$\square$