# OpenReview forum: "The Gain from Ordering in Online Learning"
_NeurIPS.cc/2023/Conference — NeurIPS 2023 poster_

### Official Review · Reviewer_5ur6 · 2023-06-08

**Soundness:** 3 good
**Presentation:** 2 fair
**Contribution:** 3 good
**Rating:** 7
**Confidence:** 4

**Summary:**

This paper studies the problem of self-directed online learning, where a set of examples (i.e., features) is provided in advance, and the learner has the autonomy to determine the order of the examples with the aim of minimizing their regret. The primary focus is on realizable linear regression and ReLU regression under square loss, assessing the ratio between the best achievable regret with random order and the achievable regret with a self-directed order (termed as the 'Gain').

The main contribution of the paper aims to highlight the discrepancy between the efficiently achievable Gain and the optimal information-theoretical Gain. The key findings are as follows:

1. If the optimal weight $w^*$ is sampled uniformly from the unit sphere, and the example set is chosen adversarially, then no polynomial time algorithm achieves the Gain of order $d^{-1/\log\log^c d}$ times the optimal information-theoretical Gain, assuming the Exponential Time Hypothesis (ETH).
2. If $w^*$ is adversarial but the example set $X$ is sampled uniformly from the unit sphere, then a computationally efficient algorithm exists that achieves the Gain of order $1/\log d$ times the optimal information-theoretical Gain.
3. Absolute lower bounds for computationally efficient Gains are also established for the Linear and ReLU regression.

*After the rebuttal: In light of the responses provided by the authors and my improved understanding of the impact of this work, I have decided to adjust my rating to "accept."*

**Strengths:**

The main strength of this paper lies in its unique presentation of a hardness result on the efficiently achievable Gain in the context of self-directed online learning. As far as I am aware, this is the first of its kind in this scenario. The reduction used in the proof is interesting and could potentially be useful in obtaining hardness results for other learning theoretical problems. In addition, the paper employs interesting geometric concepts to establish the efficiently achievable Gain in cases of random examples.

**Weaknesses:**

The primary limitation of this paper is its overall significance. While the results are intriguing from a theoretical computer science perspective, I'm not entirely convinced of their importance from a learning theoretical standpoint:

1. The significance of an impossibility result in self-directed online learning is unclear, considering that we already have efficient algorithms for linear regression, even in the adversarial case.
2. The results are somewhat challenging to fully appreciate given that the Gain is already a ratio, and the presented results are essentially a ratio of a ratio.
3. In Section 4, it would be clearer to state the results for the worst-case $w^*$ directly, rather than including it as a remark.

Nevertheless, despite these points, I believe this paper still holds interest for the NeurIPS community due to its presentation of a set of intriguing technical results. This is the primary reason for my "weak accept" recommendation.

Additionally, I noticed a couple of typographical errors:
- Line 202: A period is missing at the end of the sentence.
- Line 211: "Given" is repeated.
- Footnote of page 15 (in appendix)

**Questions:**

I have a few questions for the authors:
- Currently, the paper only considers the realizable case. Is it possible to extend the methodology to the unrealizable case (i.e., when the labels $y$ are adversarially given)?
- Is there a known result that demonstrates the distinction between random order and worst-case order in the fixed design online setup?

**Limitations:**

No issue with negative societal impact.

---

> ### Author Rebuttal · Authors · 2023-08-09
>
> ## Reviewer 5ur6
>
> We thank the reviewer for carefully reading our manuscript and providing insightful feedback. We answer the reviewer's question in the rest of the rebuttal.
>
> > The significance of an impossibility result in self-directed online learning is unclear, considering that we already have efficient algorithms for linear regression, even in the adversarial case.
>
> The significance of self-directed online learning is that: (i) it is a natural and well-motivated learning model (see also our response to reviewer eLsL) and (ii) it can potentially lead to much lower regret bounds than standard random or worst order learning.  The fact that efficient worst-order algorithms that achieve “low” (e.g., sublinear) regret exist does not mean that we shouldn’t try and make this regret even smaller by re-ordering the examples (self-directed learning).  As we discuss in the introduction of our manuscript, for $n$ examples on the real line that are classified by a linear threshold function, in worst-order  online learning $\Omega(\log n)$ mistakes are necessary. In contrast, a self-directed learner only makes one mistake.
> Our negative result shows that achieving the optimal gain from ordering (which can be very large) for arbitrary datasets is computationally intractable. In other words, even if for some dataset the (information-theoretic) optimal gain from ordering is large it is computationally intractable to find the optimal ordering of its points to achieve it.  Starting already from our work, our impossibility result motivates the study of the distribution-specific setting where structural assumptions about the distribution that generates the examples are made.
>
>
> > The results are somewhat challenging to fully appreciate given that the Gain is already a ratio, and the presented results are essentially a ratio of a ratio.
>
> In our paper, the notion of Gain is used to compare the order selected by the learner and random order. The gain itself can be seen as an absolute improvement of a self-directed learner over an online (random) learner. In our paper, we also have results that bound the Gain itself (see e.g., Proposition 4.1). However, simply considering the absolute improvement is not very meaningful from a theoretical perspective, because we have to compare the performance of a self-directed learner with the best one. For this reason, we need to introduce the term of approximate ratio, which compares a self-directed learner with the best self-directed learner. This is why our main result is presented as a ratio of ratio.  We will include this clarification in our revised manuscript.
>
>
>
> >Currently, the paper only considers the realizable case. Is it possible to extend the methodology to the unrealizable case (i.e., when the labels y  are adversarially given)?
>
> Prior to our work, nothing was known about the self-directed linear regression problem. We initiate its study focusing on the most fundamental setting of realizable linear regression. As our impossibility result reveals, even this (seemingly) vanilla setup is surprisingly difficult to fully understand.
> Studying settings where the predictions are noisy (by some adversary or at random) is a very interesting question that we plan to investigate in the future.  Currently we believe that our results could be generalized assuming mean-zero additive noise but we believe that the investigation of the effect of noise in the labels is beyond the scope of this work.
>
>
>
> > Is there a known result that demonstrates the distinction between random order and worst-case order in the fixed design online setup
>
> This is a good question. Since this is the first work on self-directed linear regression, to the best of our knowledge there is no previous result showing the distinction between a random order and the worst order. However, there is an example showing such a distinction. Assuming there are $d+m$ examples in $R^d$. The first $d$ examples are the $d$ basis vectors and the last m examples are $e_1+\epsilon_i$, where $\epsilon_i$ is a tiny noise. In this case, the worst order would be picking $e_1,e_2,\dots,e_d$, while as m increases, a random order is likely to pick examples around $e_1$. So there will be a clear separation between the worst order and the random order. We will add a section in the appendix to explain this in detail in the next revision of our paper.

---

> > ### Comment · Reviewer_5ur6 · 2023-08-14
> >
> > I thank the authors for addressing my concerns. After reading the rebuttals and the feedback from other reviewers, I have decided to maintain my current rating, favorable to accepting this paper.

---

> > > ### Author Response · Authors · 2023-08-17
> > > **Ack**
> > >
> > > We would like to thank the reviewer for carefully going over our response!

---

### Official Review · Reviewer_N13f · 2023-07-06

**Soundness:** 3 good
**Presentation:** 3 good
**Contribution:** 3 good
**Rating:** 5
**Confidence:** 1

**Summary:**

The paper considers an online regression problem and attempted to achieve the optimal gain of ordering with efficient algorithms. For general concept class, the optimal gain of ordering is $O(n^{2/d})$, where $n$ is the number of data, $d$ is the dimension of the data.

The paper first show there is not polynomial algorithms exists to achieve optimal gain of ordering if the data label is generated arbitrarily. However, if data label is uniformly generated from a unit sphere, a $\log d$-algorithm, which picks the order of the examples based on its difficulty. When concept class is chosen to be ReLu, the optimal gain of ordering is $O( \min(d,  n^{2/d} / \log d + \log n / \log d) ) $,

**Strengths:**

not capable of evaluating

**Weaknesses:**

no

**Questions:**

no

**Limitations:**

yes

---

> ### Author Rebuttal · Authors · 2023-08-09
>
> We thank the reviewer for appreciating our work.

---

### Official Review · Reviewer_Kpn9 · 2023-07-09

**Soundness:** 3 good
**Presentation:** 3 good
**Contribution:** 3 good
**Rating:** 6
**Confidence:** 2

**Summary:**

The paper focuses on the setting of "self-directed online learning", where the online learner has access to the entire set of unlabeled points $X = \{x_1,\ldots,x_n\}$ and has to choose the point in some order $x_{\sigma(1)},\ldots,x_{\sigma(n)}$ such that at time $t$, the learner has to predict the label $y^\prime_{\sigma(t)}$ for $x_{\sigma(t)}$, then observes the true label $y_{\sigma(t)}$ and suffers loss $\ell(y_{\sigma(t)}, y^\prime_{\sigma(t)})$.

In the paper, the "gain from ordering" of a self-directed algorithm is defined w.r.t random order learning (as the ratio of the total losses of the best random order algorithm and the self-directed algorithm), where the learning algorithm picks the points according to a random permutation $\sigma$ of $[n]$.The labels for the points in $X$ will of course come from some $f$ belonging to a concept class $\mathcal{C}$ and the paper considers the gain from ordering both in expectation for a random labeling $f \sim F \in \Delta(\mathcal{C})$ (average case) and also when the worst possible labelng $f \in \mathcal{C}$ is chosen by an adversary.

* The paper shows that assuming the ETH (Exponential Time Hypothesis; 3SAT cannot be solved in subexponential time), no polynomial time self-directed learning algorithm can achieve even a $d^{1/{\rm polyloglog}(d)}$ approximation of the optimal gain from ordering when $X$ is an arbitrary subset of points in $\mathbb{R}^d$, even when the concept-class is linear ($f(x) = w \cdot x$) and the target vector is sampled uniformly from $\mathbb{S}^d$.
* When $X$ is also sampled uniformly from $\mathbb{S}^d$ (spherical data), the paper proposes and analyzes an efficient greedy self-directed learner that achieves a $\log d$ approximation of the optimal gain and also shows that the gain from ordering for this algorithm is at least $\min(\Omega(d),\Omega(n^{2/d}))$.
* For the ReLU linear regression problem (labeling is $f(x) = \mathrm{Relu}(w \cdot x)$ for some $w \in \mathbb{R}^d$) with spherical data, the paper presents an efficient self-directed learner that achieves a gain of $\min(\Omega(d), \Omega(n^{2/d}/ log d + log n/ log d))$ with some non-trivial assumptions.

**Strengths:**

* The ETH-hardness result for approximating the optimal gain from ordering is very interesting from a technical perspective (relies on a sequence of reductions showing that approximating optimal gain from ordering is at least as hard as approximating the densest $k$-subgraph problem for some appropriate choice of approximation ratios).
* The reduction reveals a very interesting equivalence between optimizing the gain from ordering and choosing an ordering that optimizes the sequential spanning cost of the data points --- which is the sum of the squares of the $\ell_2$-norms of the projections of the points taken in some order, where the $i$-th point is projected to the orthogonal complement of the span of the previous $(i-1)$ points.
* The efficient self-directed algorithm for linear regression for spherical data very intuitively exploits the above equivalence, which may be of independent interest.

**Weaknesses:**

* The ETH-hardness result requires a lot of technical machinery which is mostly relegated to the supplementary material. The proof sketches provided in the main paper are not very informative when considered "on their own".
* The spherical data assumption used to design efficient algorithms for self-directed linear regression and ReLU linear regression is an interesting enough starting point theoretically, but may not be very useful in a lot of practical scenarios. Even considering random (even i.i.d) data, less-restrictive distributional assumptions would be needed.
* No experimental results or even code are provided for the proposed algorithms.

**Questions:**

* Could the authors comment on the applicability of the algorithms proposed under the spherical data assumption, even in quasi-practical scenarios (assuming very well-behaved datasets, preprocessing etc)?
   * Are there any theoretical justifications for this assumption being a useful approximation for more general cases?
   * Have the algorithms been evaluated on simulated data?

**Limitations:**

None (w.r.t negative social impact)

---

> ### Author Rebuttal · Authors · 2023-08-09
>
> ## Reviewer Kpn9
>
> > The ETH-hardness result requires a lot of technical machinery which is mostly relegated to the supplementary material. The proof sketches provided in the main paper are not very informative when considered "on their own"
>
> We thank the reviewer for pointing out this shortcoming. The proof of the hardness result is complicated and due to the space limitations, we are only able to break down the proofs into several pieces and present the key technical lemmas in the main body of the paper. We will modify the proof sketches in the main body and add a more detailed overview in the next revision of this work.
>
> > The spherical data assumption used to design efficient algorithms for self-directed linear regression and ReLU linear regression is an interesting enough starting point theoretically, but may not be very useful in a lot of practical scenarios. Even considering random (even i.i.d) data, less-restrictive distributional assumptions would be needed. Could the authors comment on the applicability of the algorithms proposed under the spherical data assumption, even in quasi-practical scenarios (assuming very well-behaved datasets, preprocessing etc)? Are there any theoretical justifications for this assumption being a useful approximation for more general cases?
>
> This is a good point. Our hardness result shows that if we consider a very general dataset, even if we know the target is drawn uniformly from a sphere, it is computationally hard to compute an order of examples that approximates the optimal order very well. That is why the assumption of the structure of the dataset is necessary to get a non-trivial approximation result. In the distribution-dependent setting, uniform distribution over a sphere (or standard Gaussian) is perhaps the most standard assumption [1,2,3].
> Our negative result shows that for arbitrary datasets it is computationally impossible to obtain (approximately) optimal orderings while our positive results show that a simple greedy heuristic is almost optimal for spherical data.  It is a very interesting direction for future work to investigate whether it is possible to obtain efficient ordering algorithms for broader (but non worst-case) structured distributions.
>
> Furthermore, our greedy algorithm for linear regression can still be applied to more general (non uniform on the sphere) datasets.  In fact, we implemented our method and in our experiments (see next question) we empirically observed that our greedy heuristic can outperform the random order for datasets that are far from being uniform on the sphere.
>
> [1] Dasgupta S, Kalai A T, Monteleoni C. Analysis of perceptron-based active learning[C]//International conference on computational learning theory. Berlin, Heidelberg: Springer Berlin Heidelberg, 2005: 249-263.
>
> [2] Diakonikolas I, Kane D M, Kontonis V, et al. Learning general halfspaces with general massart noise under the Gaussian distribution[C]//Proceedings of the 54th Annual ACM SIGACT Symposium on Theory of Computing. 2022: 874-885.
>
> [3] Yan S, Zhang C. Revisiting perceptron: Efficient and label-optimal learning of halfspaces[J]. Advances in Neural Information Processing Systems, 2017, 30.
>
> > No experimental results or even code are provided for the proposed algorithms. Have the algorithms been evaluated on simulated data?
>
> Even though our primary goal in this work is to establish a theoretical understanding of this problem, we agree with the reviewer that implementing our method and evaluating it empirically is a nice addition to our manuscript. We implemented our greedy algorithm (see Algorithm 1) and tested it using datasets drawn uniformly on the unit sphere spherical dataset and also data generated by mixtures of spherical data (similar to Gaussian mixtures).  We observe that our simple greedy method is computationally efficient (and can be implemented with a few lines of code) and also consistently outperforms the random-order method (see attached figures). We also implemented our algorithm for ReLU regression (Algorithm 2) over a spherical dataset. We see our algorithm achieves a much smaller regret than that of a random order. Furthermore, we notice that if our algorithm is given a suitable initialization, then it has an even much better performance. More comprehensive empirical evaluation on the gain of ordering in regression (on non-synthetic data) is an interesting question for future work.
>
> We state the details of our experiments here. The first set of experiments is to evaluate the performance of Algorithm 1 over a spherical dataset. For a single experiment, we sample $20d$ examples and a target vector $w$ uniformly from the unit sphere in $R^d$. We label the dataset with an order computed by Algorithm 1 and a random order, then we compute the total regret respectively. We perform the experiment for $d=5,10,\dots,75$, for each $d$, we repeat the experiment 30 times and compute the average regret. The result of this set of experiments is shown in the first attached figure.
>
> The second set of experiments is to evaluate the performance of Algorithm 2 over the spherical dataset. The data generation process is the same as the one in the first set of experiments. However, in this set of experiments, we will compare three different settings. 1. ReLU regression with a random order. 2. Algorithm 2 without an initialization. 3. Algorithm 2 with initialization $(v,v\cdot w)$, where $v=sgn(w)$.
>
> The third set of experiments is to evaluate the performance of Algorithm 1 over more general datasets. The setting of the experiments is similar to the first set of experiments. However, the dataset is generated as follows. For each $i \in [d]$, we generate 20 examples $e_i+0.05\epsilon$, where $e_i$ is the $i$th standard basis vector and $\epsilon$ is a vector uniformly drawn from the unit sphere. Finally, we normalize every example so that each of them has a unit norm. The result of this experiment is shown in the third attached figure.

---

> > ### Comment · Reviewer_Kpn9 · 2023-08-21
> > **Response**
> >
> > Thank you very much for your response and clarifications. Improving the proof sketches in the final version by rewriting/including some more detail (as possible) would be very good.
> >
> > I do understand and agree that studying the spherical/Gaussian case is perhaps the best starting point from a theoretical perspective.
> >
> > The experimental results are also interesting, especially the practical improvement over random ordering for mixture data (non spherical).
> >
> > Given all these points and considering that some of these could be incorporated into the final version, I am willing to increase my score from 5 to 6.

---

### Official Review · Reviewer_eLsL · 2023-07-20

**Soundness:** 4 excellent
**Presentation:** 3 good
**Contribution:** 2 fair
**Rating:** 6
**Confidence:** 3

**Summary:**

The paper considers online (realizable) linear regression, in a setting where the learner may choose the order of datapoints (=self directed learning) with the goal to minimize her regret. Roughly, the gain from ordering of a self directed learner $\mathcal A$ is defined as the ratio between the best performance given random order samples, and the performance of $\mathcal A$. The paper proves that:
1. Under the exponential time hypothesis (ETH), no efficient algorithm can approximate the optimal gain to within a factor of $d^{1/\log\log^c d}$ for some constant $c$. This holds in particular in the case that the ground truth is uniform over the unit sphere.
2. Assuming that the datapoints are drawn uniformly over the unit sphere, as well as the ground truth hypothesis vector, a certain greedy algorithm (Algorithm 1) provides a $\log d$-approximation to the optimal gain.


**Strengths:**

The paper is well written, the proofs are non trivial, and from a purely theoretical perspective the question studied seems interesting,


**Weaknesses:**

The motivation for studying regret of a self directed learner is not entirely clear to me. Curriculum learning sounds like a good motivation for sample complexity or convergence rate, but not regret. In which truly online setting would the learner have access to the order of datapoints?
To me this is really the main weakness, I do not understand the motivation of this question in the context of machine learning problems.

**Questions:**

### Minor comments
- Lines 62-63 are repeated inside Remark 1.3
- Line 84: "one mistake suffices" - suffices for what?
- Theorem 1.5: "Let X be an arbitrary set..."; There exists sets (e.g., singletone X) for which it is easy to approximate the optimal gain. I assume the claim should refer the (non) existence of an algorithm that works for arbitrary sets.
- When using 99% in the theorem statements it is unclear what is the dependence on high probability factor.
- Notation for the datapoints is inconsistent sometimes, for example in Definition 3.3, first $x_i$ then $x^{(\sigma(i))}$
- In Algorithm 1, I think $L_{i-1}$ in the selection of $x^{(i)}$ should be $L_{i-1}^\perp$.

---

> ### Author Rebuttal · Authors · 2023-08-09
>
>
> ## Reviewer eLsL
>
> >The motivation for studying the regret of a self-directed learner is not entirely clear to me. Curriculum learning sounds like a good motivation for sample complexity or convergence rate, but not regret. In which truly online setting would the learner have access to the order of data points? To me this is really the main weakness, I do not understand the motivation of this question in the context of machine learning problems.
>
> We thank the reviewer for carefully reading our manuscript and providing insightful feedback. Self-directed learning is a fundamental online learning model that studies if the ability to reorder examples can help a learner make better predictions. The study of self-directed learning can be traced back to the 1900s e.g. [1,2,3] and even very recently there is still surprising progress made in this direction e.g. [4], which shows in the problem of online linear classification if the learner has the power of reordering then the number of mistakes made by the learner to learn the target linear threshold function is significantly smaller than that in the setting where the order is chosen by an adversary. Our work extends the classification setting studied in the literature to the more general regression setting.
>
> In practice, there are also many direct applications of this learning model. A direct application is directed marketing [5,6], one of the common and crucial business intelligence tasks, which is a process of identifying likely buyers to market products accordingly. In direct marketing, an agent must study customers’ characteristics and needs, and select customers to market its products to minimize its market loss. As a more concrete example, a streaming service or social media platform may want to learn the content preferences of customers without making too many bad recommendations. In this application, the platform chooses the order in which to present the content (from a pool of available videos) in order to minimize the “regret” (and keep the user engaged).  We will include a more detailed discussion on motivation in our revised manuscript.
>
>
> [1] Ben-David S, Kushilevitz E, Mansour Y. Online learning versus offline learning[J]. Machine Learning, 1997, 29: 45-63.
>
> [2] Goldman S A, Sloan R H. The power of self-directed learning[J]. Machine Learning, 1994, 14: 271-294.
>
> [3] Ben-David S, Eiron N. Self-directed learning and its relation to the VC-dimension and to teacher-directed learning[J]. Machine Learning, 1998, 33: 87-104.
>
> [4] Diakonikolas I, Kontonis V, Tzamos C, et al. Self-Directed Linear Classification[C]//The Thirty Sixth Annual Conference on Learning Theory. PMLR, 2023: 2919-2947.
>
> [5] Ling, Charles X., and Chenghui Li. "Data mining for direct marketing: Problems and solutions." Kdd. Vol. 98. 1998.
>
> [6] Ni, Eileen A., and Charles X. Ling. "Direct marketing with fewer mistakes." Advanced Data Mining and Applications: 7th International Conference, ADMA 2011, Beijing, China, December 17-19, 2011, Proceedings, Part I 7. Springer Berlin Heidelberg, 2011.
>
> Minor Comments
> We thank the reviewer for pointing out typos and making suggestions for improving our manuscript. Here are answers to some of the minor comments.
> - Line 84 => one mistake suffices to learn the one dimension threshold function.
> - We agree with the reviewer and will restate Theorem 1.5 so that it is more clear.
> - The randomness for spherical data only comes from the fact that the points of the dataset $X$ are drawn i.i.d. from the uniform distribution on the unit sphere.

---

> > ### Comment · Reviewer_eLsL · 2023-08-15
> >
> > Thank you for your rebuttal.
> >
> > My main concern has been addressed, and I have decided to raise my score.

---

### Author Rebuttal · Authors · 2023-08-09

We want to thank all reviewers for taking the time to read our manuscript carefully and for providing constructive and insightful feedback. We are very encouraged by the positive comments of the reviewers on the **novelty of the ETH hardness result** (Reviewers eLsL,Kpn9, 5ur6), **interesting and potentially useful techniques for proving the hardness result**(Reviewers Kpn9, 5ur6), **geometric intuitiveness of the self-directed learning algorithms** (Reviewers Kpn9, 5ur6), **the writing quality and the clarity of the presentation of the ideas** (Reviewer eLsL).

We attach a pdf file here that contains a numerical experiment that we did and provide detailed responses to each reviewer separately. We look forward to engaging in further discussion with the reviewers, answering questions, and discussing improvements.

---

### Decision · Program_Chairs · 2023-09-21

**Decision:**

Accept (poster)

**Comment:**

The paper examines online linear regression, where a learner can order data points to minimize regret. The paper proves that, under the exponential time hypothesis (ETH), no algorithm can efficiently approximate optimal gain A greedy algorithm provides an approximation to optimal gain assuming uniform data and ground truth. Reviewers found the paper theoretically interesting but questioned the motivation. The author provided a convincing argument in favor, citing historical and recent advancements, and they also offer examples like directed marketing and content recommendation systems.